# Physics-Informed Distillation of Diffusion Models for PDE-Constrained Generation

Yi Zhang [1]  Peng Wang [2]  Difan Zou [3]

## Abstract

Diffusion models show growing promise for generative modeling of physical systems, but enforcing partial differential equation (PDE) constraints directly is infeasible during the stochastic denoising process. Current methods apply constraints to the expected clean sample, incurring a *Jensen's Gap* that forces a trade-off between PDE satisfaction and generative accuracy. To bridge this gap, we propose **P**hysics-**I**nformed **D**istillation of **D**iffusion **M**odels (PIDDM), a simple yet effective post-hoc distillation strategy that enforces PDE constraints after training. PIDDM enables fast single-step generation while improving both physical consistency and sample quality, supporting forward/inverse problems and reconstruction from partial observations. Extensive experiments across PDE benchmarks show PIDDM outperforms recent baselines, such as PIDM (Bastek et al., 2025), DiffusionPDE (Huang et al., 2024a), and ECI-sampling (Cheng et al., 2025), in both accuracy and constraint satisfaction, with lower computation and minimal hyperparameter tuning, offering a more efficient pathway to physics-informed diffusion models.

## 1. Introduction

Solving partial differential equations (PDEs) underpins innumerable applications in physics, biology, and engineering, spanning fluid flow (Davidson, 2015), heat transfer (Incropera et al., 2011), elasticity (Timoshenko & Goodier, 1970), electromagnetism (Jackson, 1998), and chemical diffusion (Crank, 1975). Classical discretization schemes such as finite-difference (Smith, 1985) and finite-element

methods (LeVeque, 2007) provide reliable solutions, but their computational cost grows sharply with mesh resolution, dimensionality, and parameter sweeps, limiting their practicality for large-scale or real-time simulations (Hughes, 2000). This bottleneck has fueled a surge of learning-based solvers that approximate or accelerate PDE solutions, from early physics-informed neural networks (PINNs) (Raissi et al., 2019) to modern operator-learning frameworks such as DeepONet (Lu et al., 2019), Fourier Neural Operators (Li et al., 2020) and Physics-Informed Neural Operator (Li et al., 2024), offering faster inference, uncertainty quantification, and seamless integration into inverse or data-driven tasks.

Among these learning-based solvers, diffusion models (Ho et al., 2020; Song et al., 2020) provide a promising framework for generative modeling of physical systems. For PDEs, a diffusion model can learn the joint distribution of solution and coefficient fields, $x_0 = (u, a)$, from data, where $a$ denotes input parameters that satisfy the boundary operator $\mathcal{B}$ (for example, material properties or initial conditions) and $u$ is the corresponding solution that satisfies the PDE operator $\mathcal{F}$. After training, the model can sample $(u, a)$ from this learned distribution, enabling forward simulation (sample $u$ given $a$), inverse recovery (sample $a$ given $u$), and conditional reconstruction (complete missing components of $u$ or $a$) *within a single framework, which prior non-diffusion approaches (Li et al., 2020; Lu et al., 2019; Raissi et al., 2019) do not provide.* However, while diffusion models perform well under soft, high-level constraints (Rombach et al., 2022; Esser et al., 2024; Ho & Salimans, 2022; Chung et al., 2023; Ben-Hamu et al., 2024), PDE applications often require strict, low-level constraints dictated by $\mathcal{F}$ and $\mathcal{B}$.

Enforcing such PDE constraints within diffusion models is nontrivial. A core difficulty is that, at an individual noise level $t$, diffusion models operate on noisy variables $x_t$ rather than the clean physical field $x_0$, where constraints such as $\mathcal{F}[x_0] = 0$ are defined. To address this, one option is to reconstruct $x_0$ by running the full deterministic sampling trajectory, but this is computationally expensive since it requires many forward passes, and enforcing constraints through backpropagation often causes gradient issues (Bastek et al., 2025). A more common alternative is to

[1]Institute of Data Science, The University of Hong Kong [2]Department of Computer and Information Science, University of Macau [3]Institute of Data Science & School of Computing and Data Science, The University of Hong Kong. Correspondence to: Difan Zou <dzou@cs.hku.hk>.

*Proceedings of the 43rd International Conference on Machine Learning*, Seoul, South Korea. PMLR 306, 2026. Copyright 2026 by the author(s).

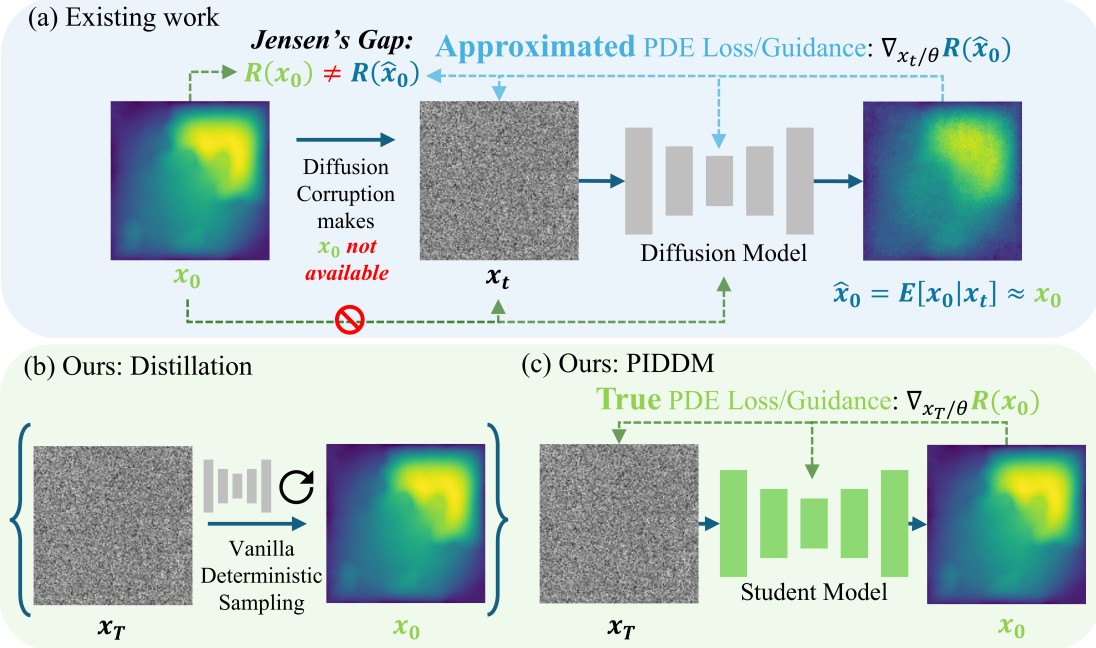

*Figure 1.* **Illustration of physics-constrained diffusion generation and our proposed framework.** **(a)** Existing methods (Huang et al., 2024a; Cheng et al., 2025; Bastek et al., 2025; Jacobsen et al., 2024) impose PDE losses or guidance on the posterior mean $\mathbb{E}[x_0|x_t]$ in diffusion training and sampling, introducing the Jensen's Gap. **(b)** We train and sample a diffusion model using vanilla methods to generate a paired noise-data dataset for distillation. **(c)** Our proposed framework distills the teacher diffusion model and directly enforces physical constraints on the final generated sample $x_0$, avoiding posterior-mean surrogate constraints in the student objective.

approximate $\boldsymbol{x}_0$ with the posterior mean $\mathbb{E}[\boldsymbol{x}_0|\boldsymbol{x}_t]$, which can be efficiently computed via Tweedie's formula (Bastek et al., 2025; Huang et al., 2024a; Cheng et al., 2025; Jacobsen et al., 2024; Kawar et al., 2022; Wang et al., 2022; Zhu et al., 2023; Zhang et al., 2025; Song et al., 2023a; Tang et al., 2024; Huang et al., 2024b) (see the right panel of Fig. 1(a)). However, this introduces a mismatch: enforcing constraints on the posterior mean, $\mathcal{F}[\mathbb{E}[\boldsymbol{x}_0|\boldsymbol{x}_t]]$, is not equivalent to enforcing the expected constraint, $\mathbb{E}[\mathcal{F}[\boldsymbol{x}_0]|\boldsymbol{x}_t]$, due to Jensen's inequality. This issue, known as the *Jensen's Gap* in inverse-problem diffusion guidance (Chung et al., 2023) and later studied in PDE-constrained diffusion models (Bastek et al., 2025; Huang et al., 2024a), can lead to degraded physical fidelity.

**Contributions.** We propose an effective framework that enforces PDE constraints in diffusion models via post-hoc distillation, enabling reliable and efficient generation under physical laws. As shown in Fig. 1(c), our method sidesteps the limitations of existing constraint-guided diffusion-based approaches by decoupling physics enforcement from the diffusion trajectory. Our main contributions are:

- **Empirical confirmation of Jensen's Gap:** We provide an empirical demonstration and quantitative analysis of the *Jensen's Gap* in PDE-constrained diffusion, a discrepancy that arises when PDE constraints are imposed on the posterior mean $\mathbb{E}[\boldsymbol{x}_0|\boldsymbol{x}_t]$, rather than the final clean

sample $\boldsymbol{x}_0$.

- **Final-sample physics enforcement:** PIDDM changes where and when physics is enforced: instead of guiding iterative denoising through posterior-mean surrogates, it imposes PDE supervision on the final output of a post-hoc distilled student. This removes the posterior-mean versus final-sample mismatch from the student objective, while our distributional fidelity claims remain empirical.

- **Versatile and efficient inference:** The distilled student model preserves the full generative capabilities of the teacher, supporting physical simulation, reconstruction, and unified forward and inverse PDE solving within *a single model*, while enabling one-step generation for fast inference. Experiments across diverse PDEs show that PIDDM surpasses posterior-mean-based methods (Huang et al., 2024a; Cheng et al., 2025; Bastek et al., 2025; Jacobsen et al., 2024) in both generation quality and constraint satisfaction.

## 2. Problem Setup: Jensen's Gap in Diffusion Models with PDE Constraints

In scientific machine learning, there exist many *hard* and *low-level* constraints that are mathematically strict and non-negotiable (LeVeque & Leveque, 1992; Hansen et al., 2023; Mouli et al., 2024; Saad et al., 2022). In this section, we discuss how existing works impose these constraints in

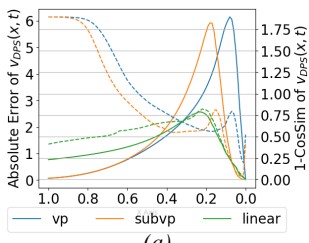 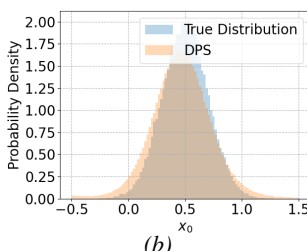 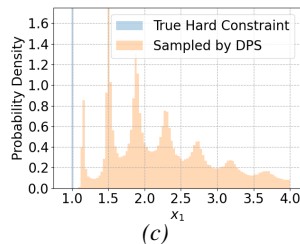 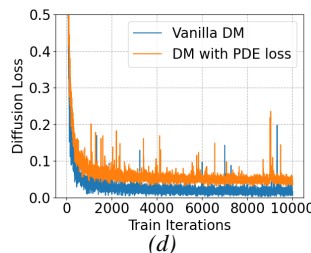

*Figure 2.* **Empirical illustration of the Jensen's Gap in physics-constrained diffusion models.** **(a)** Absolute velocity error and angular discrepancy $(1 - \cos(\theta))$ between Diffusion Posterior Sampling (DPS) and the ground-truth conditional ODE velocity on the MoG dataset. **(b)** and **(c)** Histograms comparing the first (unconstrained) and second (hard-constrained) dimensions of DPS-sampled MoG data against the ground-truth MoG. **(d)** Training-time manifestation: diffusion loss comparison on a Stokes problem dataset.

diffusion-generated data, and the Jensen's Gap introduced by posterior-mean guidance in inverse problems (Chung et al., 2023) and inherited by PDE-constrained diffusion methods (Bastek et al., 2025; Huang et al., 2024a).

### 2.1. Preliminaries on Physical Constraints

Physical constraints are typically expressed as a *partial differential equation (PDE)* $\mathcal{F}$ defined over a solution domain $\Omega \subset \mathbb{R}^d$, together with a *boundary condition* operator $\mathcal{B}$ defined on the coefficient domain $\Omega'$:

$$\mathcal{F}[\mathbf{u}(\boldsymbol{\xi})] = 0 \text{ for } \boldsymbol{\xi} \in \Omega, \quad \mathcal{B}[\mathbf{a}(\boldsymbol{\xi}')] = 0 \text{ for } \boldsymbol{\xi}' \in \Omega'. \quad (1)$$

In practice, the domains $\Omega$ and $\Omega'$ are discretized into a uniform grid, typically of size $H \times W$, and the fields $\boldsymbol{u}$ and $\boldsymbol{a}$ are evaluated at those grid points to produce the observed data $\boldsymbol{x}_0 = (\boldsymbol{u}, \boldsymbol{a})$, where diffusion models are trained to learn the joint distribution $p(\boldsymbol{x}_0) = p((\boldsymbol{u}, \boldsymbol{a}))$. While PINNs (Raissi et al., 2019) model the mapping $\boldsymbol{\xi}, \boldsymbol{\xi}' \mapsto (\mathbf{u}(\boldsymbol{\xi}), \mathbf{a}(\boldsymbol{\xi}'))$ with differentiable neural networks to enable automatic differentiation (Paszke, 2019; Abadi et al., 2016), grid-based approaches commonly approximate the differential operators in $\mathcal{F}$ via finite difference methods (Smith, 1985; LeVeque, 2007). To quantify the extent to which a generated sample $\boldsymbol{x}_0$ violates the physical constraints, the *physics residual error* is often defined by:

$$\mathcal{R}(\boldsymbol{x}_0) = \mathcal{R}((\boldsymbol{u}, \boldsymbol{a})) := \begin{bmatrix} \mathcal{F}[\boldsymbol{u}], \mathcal{B}[\boldsymbol{a}] \end{bmatrix}^\top \quad (2)$$

Here, $\mathcal{R}(\boldsymbol{x})$ measures the discrepancy between the sample $\boldsymbol{x}$ and the expected PDE $\mathcal{F}$ and boundary conditions $\mathcal{B}$. The *physics residual loss* is often defined by the squared norm of this physics residual error, i.e., $\|\mathcal{R}(\boldsymbol{x})\|^2$.

### 2.2. Imposing PDE Constraints in Diffusion Models

The physical constraints $\mathcal{R}$ are often defined on the clean field $\boldsymbol{x}_0$, while during training or sampling of the diffusion model, the model only observes the noisy state $\boldsymbol{x}_t$. Therefore, direct optimization or controlled generation based on the physical residual loss $\mathcal{R}(\boldsymbol{x}_0)$ is generally impractical.

A practical workaround is to evaluate the constraint on an estimate of $\boldsymbol{x}_0$ from $\boldsymbol{x}_t$, and a common choice is to use the estimated *posterior mean*: $\mathbb{E}[\boldsymbol{x}_0 \mid \boldsymbol{x}_t]$ based on the score network in a diffusion model (Bastek et al., 2025; Huang et al., 2024a). As a simplified example, consider the forward process defined as $\boldsymbol{x}_t = \boldsymbol{x}_0 + \sigma_t \boldsymbol{\epsilon}$, where $\sigma_t$ denotes the noise level at time $t$ and $\boldsymbol{\epsilon} \sim \mathcal{N}(0, \mathbf{I})$ is standard Gaussian noise. Then, the posterior mean can be efficiently estimated via Tweedie's formula:

$$\hat{\boldsymbol{x}}_\theta(\boldsymbol{x}_t, t) := \boldsymbol{x}_t + \sigma_t^2 s_\theta(\boldsymbol{x}_t, t)$$
$$\approx \boldsymbol{x}_t + \sigma_t^2 \nabla \log p(\boldsymbol{x}_t) \approx \mathbb{E}[\boldsymbol{x}_0 \mid \boldsymbol{x}_t],$$

where $s_\theta$ is a learned score function approximating the gradient of the log-density (see Appendix B.2 for the derivation for the general diffusion process). Leveraging this approximation, several existing works incorporate PDE constraints by evaluating the PDE residual operator $\mathcal{R}$ on $\hat{\boldsymbol{x}}_\theta(\boldsymbol{x}_t, t)$. For instance, **PIDM** (Bastek et al., 2025) integrates PDE constraints into a diffusion model *at training time* by augmenting the standard diffusion objective with an additional PDE residual loss $\mathcal{R}(\hat{\boldsymbol{x}}_\theta(\boldsymbol{x}_t, t))$. Similarly, *at inference time*, **DiffusionPDE** (Huang et al., 2024a) and **CoCoGen** (Jacobsen et al., 2024) employ diffusion posterior sampling (DPS) (Chung et al., 2023), guiding each intermediate sample $\boldsymbol{x}_t$ using the gradient $\nabla_{\boldsymbol{x}} \mathcal{R}(\hat{\boldsymbol{x}}_\theta(\boldsymbol{x}_t, t))$. On the other hand, **ECI-sampling** (Cheng et al., 2025) directly applies hard constraints to the posterior mean at each DDIM step using a correction operator (more detailed discussion on their implementations can be found in Appendix D.4). Beyond PDE applications, constrained diffusion models for image inverse problems also rely on posterior-mean approximations for true posterior estimation. Representative examples include DDRM (Kawar et al., 2022), DDNM (Wang et al., 2022), LGD (Song et al., 2023a), DPG (Tang et al., 2024), SCG (Huang et al., 2024b), DCDPM (Dong et al., 2024), mid-point guidance (Moufad et al., 2024), DiffPIR (Zhu et al., 2023), and DAPS (Zhang et al., 2025) (see Appendix A.2 for details). While these pioneering methods have been demonstrated to be effective in enforcing PDE constraints within diffusion models, they still suffer a theo-

retical inconsistency: PDE constraints are enforced on the posterior mean approximation $\mathbb{E}[\boldsymbol{x}_0|\boldsymbol{x}_t]$, which is not equivalent to the constraints on the true generated data $\boldsymbol{x}_0$ due to Jensen's inequality:

$$\mathcal{R}(\mathbb{E}[\boldsymbol{x}_0|\boldsymbol{x}_t]) \neq \mathbb{E}[\mathcal{R}(\boldsymbol{x}_0)|\boldsymbol{x}_t]. \qquad (3)$$

This discrepancy is commonly referred to as the *Jensen's Gap* (Chung et al., 2023; Bastek et al., 2025; Huang et al., 2024a; Gao et al., 2017). To mitigate this issue, PIDM and DiffusionPDE heuristically down-weight PDE constraints at early denoising steps (large $t$) in training and sampling, respectively, where the gap is pronounced, and emphasize them near $t \to 0$, where the posterior mean approximation improves. ECI-sampling introduces stochastic resampling steps (Wang et al., 2025) to project intermediate samples back toward their correct distribution. Many other methods for image inverse problems also provide partial improvements to reduce approximation error (Song et al., 2023a; Tang et al., 2024; Huang et al., 2024b; Dong et al., 2024; Moufad et al., 2024) (Appendix A.2). These methods address aspects of the approximation error, but still rely on posterior-mean surrogates during the diffusion trajectory.

## 2.3. Demonstration of the Jensen's Gap

To better illustrate this posterior-mean mismatch and its negative effect in PDE-constrained generation, we conduct experiments on two synthetic datasets: a Mixture-of-Gaussians (MoG) dataset and a Stokes Problem dataset.

**Sampling-time Jensen's Gap.** We demonstrate the sampling-time Jensen's Gap using the Mixture-of-Gaussians (MoG) dataset, where the score function is analytically tractable, allowing us to isolate the effect of the diffusion process without interference from training error. The MoG is constructed in 2D: the first dimension follows a bimodal Gaussian distribution, while the second dimension encodes a discrete latent variable that serves as a hard constraint. Concretely, the joint distribution is defined as a mixture of two Gaussians, each supported on a distinct horizontal line:

$$p(x_0) = 0.5 \cdot \mathcal{N}(x_1; -1, \sigma^2) \cdot \delta(x_2 + 1) +$$
$$0.5 \cdot \mathcal{N}(x_1; +1, \sigma^2) \cdot \delta(x_2 - 1),$$

where $\delta(\cdot)$ denotes the Dirac delta function and $\sigma = 0.2$. To examine the impact of Jensen's Gap during sampling, we compare Diffusion Posterior Sampling (DPS) (Chung et al., 2023) which uses a latent code to guide the generation, with the ground-truth conditional ODE trajectory derived analytically. We evaluate three representative diffusion processes: Variance-Preserving (VP) (Ho et al., 2020), Sub-VP (Song et al., 2020), and Linear (Liu et al., 2023a), and compare their velocity fields during inference to characterize Jensen's Gap. To quantify amplitude errors, we compute the mean absolute error (MAE) and angular error between

the DPS-predicted velocity field $v_{\text{DPS}}(x, t)$ and the ground-truth velocity $v_{\text{GT}}(x, t)$: We observe that both DPS errors are significantly elevated at intermediate timesteps and only diminish as $t \to 0$, as shown in Fig. 2a. Although DPS achieves accurate sampling in the unconstrained dimension (Fig. 2b), it fails to respect the hard constraint in the constrained dimension (Fig. 2c). This experiment illustrates the mismatch induced by posterior-mean guidance; it does not claim that single posterior samples along the trajectory resolve the gap.

**Training-time Jensen's Gap.** We examine the Jensen's Gap during training using the synthetic Stokes dataset with target distribution $p_{\text{Stokes}}$. The diffusion model $\boldsymbol{v}_\theta$ adopts a Fourier Neural Operator (FNO) (Li et al., 2020) architecture and follows a standard linear noise schedule (Lipman et al., 2023; Liu et al., 2023b;c). Dataset and training details are given in Appendices C and D. We take PIDM (Bastek et al., 2025) as a representative method, which augments the diffusion loss with a PDE residual term $\mathcal{R}(\hat{\boldsymbol{x}}_\theta(\boldsymbol{x}_t, t))$, and compare its performance with standard diffusion training. To assess generative quality, we monitor the diffusion loss, which theoretically corresponds to the evidence lower bound (ELBO) (Ho et al., 2020; Kingma et al., 2013). The comparison results are shown in Fig. 2d, revealing a significant increase in diffusion loss when the PDE residual loss is incorporated. This suggests that the PDE residual loss does not improve modeling of the PDE-consistent data distribution. This observation also corroborates findings from PIDM (Bastek et al., 2025), which identified that residual supervision on the posterior mean can create "a conflicting objective between the data and residual loss", where the data loss represents the original diffusion training objective. These results provide further evidence of the posterior-mean mismatch in training, as enforcing constraints on $\mathbb{E}[\boldsymbol{x}_0|\boldsymbol{x}_t]$ may interfere with maximizing the likelihood of the true data distribution.

## 3. Method: Physics-Informed Distillation of Diffusion Models

In Section 2, we have demonstrated the posterior-mean mismatch that appears when incorporating physical constraints into diffusion training and sampling, as observed in prior works. To address this issue, we propose a distillation-based framework that sidesteps this mismatch in the student objective. Specifically, instead of enforcing constraints on the posterior mean during the diffusion process, we apply physical constraints directly to the final generated samples in a post-hoc distillation stage.

### 3.1. Diffusion Training

To decouple physical constraint enforcement from the diffusion process itself, we first conduct *standard* diffusion

**Algorithm 1** PIDDM Training: Physics-Informed Distillation.

**Require:** Teacher Model $\boldsymbol{v}_\theta(\boldsymbol{x}, t)$, Student Model $d_{\boldsymbol{\theta}'}$, Batch Size $B$, Terminal Time $T{=}1$, Steps $N_s$, Step Size $\mathrm{d}t{=}1/N_s$, Physics Residual Error $\mathcal{R}$, Loss Weight $\lambda_{\text{train}}$, Learning Rate $\eta_{\text{train}}$

1: **repeat**
2:     Sample $\boldsymbol{\epsilon}_{1:B} \overset{\text{i.i.d.}}{\sim} \mathcal{N}(\mathbf{0}, \mathbf{I})$;    $\boldsymbol{x}_T \leftarrow \boldsymbol{\epsilon}_{1:B}$
3:     **for** $t = T{-}\mathrm{d}t, \dots, 0$ **do**
4:         $\boldsymbol{x}_t \leftarrow \boldsymbol{x}_{t+\mathrm{d}t} - \boldsymbol{v}_\theta(\boldsymbol{x}_{t+\mathrm{d}t}, t{+}\mathrm{d}t)\,\mathrm{d}t$ ▷ **Sampling Phase**
5:     **end for**
6:     $\boldsymbol{x}_{\text{pred}} \leftarrow d_{\boldsymbol{\theta}'}(\boldsymbol{\epsilon}_{1:B})$
7:     $\mathcal{L} \leftarrow \frac{1}{B}(\|\boldsymbol{x}_{\text{pred}} - \boldsymbol{x}_0\|^2 + \lambda_{\text{train}}\|\mathcal{R}(\boldsymbol{x}_{\text{pred}})\|^2)$    ▷ **Distillation Phase**
8:     $\theta' \leftarrow \theta' - \eta_{\text{train}}\nabla_{\theta'}\mathcal{L}$
9: **until** Converged

---

**Algorithm 2** PIDDM Inference: Physics Data Simulation

1: **Input** Student Model $d_{\boldsymbol{\theta}'}$, Physics Residual Error $\mathcal{R}$, Refinement Step Number $N_f$, Refinement Step Size $\eta_{\text{ref}}$, Latent Noise $\boldsymbol{\epsilon} \sim \mathcal{N}(\mathbf{0}, \mathbf{I})$.
2: **for** $i = 1, \dots, N_f$ **do**
3:     $\boldsymbol{\epsilon} \leftarrow \boldsymbol{\epsilon} - \eta_{\text{ref}}\nabla_{\boldsymbol{\epsilon}}\|\mathcal{R}(d_{\boldsymbol{\theta}'}(\boldsymbol{\epsilon}))\|^2$
4: **end for**
5: **Output** $d_{\boldsymbol{\theta}'}(\boldsymbol{\epsilon})$

---

**Algorithm 3** PIDDM Inference for Forward/Inverse/Reconstruction

1: **Input** Student Model $d_{\boldsymbol{\theta}'}$, Physics Residual Error $\mathcal{R}$, Optimization Iterations $N_o$, Step Size $\eta_{\text{infer}}$, Observation $\boldsymbol{x}'$, Observation Mask $M$, Loss Weight $\lambda_{\text{infer}}$, Latent Noise $\boldsymbol{\epsilon} \sim \mathcal{N}(\mathbf{0}, \mathbf{I})$.
2: **for** $i = 1, \dots, N_o$ **do**
3:     $\boldsymbol{x}_{\text{mix}} \leftarrow \boldsymbol{x}' \odot M + d_{\boldsymbol{\theta}'}(\boldsymbol{\epsilon}) \odot (1 - M)$
4:     $\boldsymbol{\epsilon} \leftarrow \boldsymbol{\epsilon} - \eta_{\text{infer}}\nabla_{\boldsymbol{\epsilon}}[\|(d_{\boldsymbol{\theta}'}(\boldsymbol{\epsilon}) - \boldsymbol{x}') \odot M\|^2 + \lambda_{\text{infer}}\|\mathcal{R}(\boldsymbol{x}_{\text{mix}})\|^2]$
5: **end for**
6: $\boldsymbol{x} \leftarrow \boldsymbol{x}' \odot M + d_{\boldsymbol{\theta}'}(\boldsymbol{\epsilon}) \odot (1 - M)$
7: **Output** $\boldsymbol{x}$

---

model training using its original denoising objective, without adding any constraint-based loss. To obtain smoother sampling trajectories that benefit later noise-data distillation (Liu et al., 2023b;c), we adopt a linear diffusion process and apply the $\boldsymbol{v}$-prediction parameterization (Liu et al., 2023b; Lipman et al., 2023; Liu et al., 2023c; Cheng et al., 2025; Esser et al., 2024), which is commonly referred to as a flow model. Specifically, the training objective is

$$\mathcal{L}(\boldsymbol{\theta}) = \mathbb{E}_{\substack{t\sim U(0,1),\, \boldsymbol{x}_0\sim p \\ \boldsymbol{\epsilon}\sim\mathcal{N}(\mathbf{0},\boldsymbol{I})}} \left[\|\boldsymbol{v}_{\boldsymbol{\theta}}(\boldsymbol{x}_t, t) - (\boldsymbol{\epsilon} - \boldsymbol{x}_0)\|^2\right],$$

where $\boldsymbol{x}_t = (1-t)\boldsymbol{x}_0 + t\boldsymbol{\epsilon}$, $p(\boldsymbol{x}_0)$ is the distribution of joint data containing both solution and coefficient fields $\boldsymbol{x} = (\boldsymbol{u}, \boldsymbol{a})$, $\boldsymbol{\epsilon}$ is sampled from a standard Gaussian distribution, and $\boldsymbol{v}_{\boldsymbol{\theta}}$ is the neural network used as the diffusion model. This formulation allows the model to learn to reverse the diffusion process without entangling it with physical supervision, thereby preserving generative fidelity.

### 3.2. Imposing PDE Constraints in Distillation

After training the teacher diffusion model using the standard denoising objective, we proceed to the distillation stage, where we transfer its knowledge to a student model designed for efficient one-step generation. Crucially, this post-hoc dis-

tillation stage is where we impose PDE constraints, moving physics enforcement away from posterior-mean surrogates used during diffusion training or sampling. This distillation process is guided by two complementary objectives: (1) learning to map a noise sample to the final generated output predicted by the teacher model, and (2) enforcing physical consistency on this output via PDE residual minimization. Concretely, we begin by sampling a noise input $\boldsymbol{\epsilon} \sim \mathcal{N}(\mathbf{0}, \boldsymbol{I})$ and generating a target sample $x_0$ using the pre-trained teacher model via deterministic integration of the reverse-time ODE:

$$\boldsymbol{x}_{t-\mathrm{d}t} = \boldsymbol{x}_t - \boldsymbol{v}_{\boldsymbol{\theta}}(\boldsymbol{x}_t, t)\,\mathrm{d}t, \tag{4}$$

which proceeds from $t = 1$ to $t = 0$ using a fixed step size $dt$. This yields a paired noise-data dataset $\mathcal{D} = \{\boldsymbol{\epsilon}, \boldsymbol{x}_0\}$ for distillation, as shown in Fig. 1(b). A student model $d_{\boldsymbol{\theta}'}(\boldsymbol{\epsilon})$ is then trained to predict $\boldsymbol{x}_0$ in one step, as shown in Fig. 1(c). Meanwhile, to enforce physical consistency, we evaluate the physics residual error on the output $\boldsymbol{x} = d_{\boldsymbol{\theta}'}(\boldsymbol{\epsilon})$, i.e., $\|\mathcal{R}(\boldsymbol{x})\|^2$. The overall training objective is:

$$\begin{aligned}\mathcal{L}_{\text{total}} &= \mathcal{L}_{\text{sample}} + \lambda_{\text{train}}\mathcal{L}_{\text{PDE}} \\ &= \mathbb{E}_{(\boldsymbol{\epsilon},\boldsymbol{x}_0)\sim\mathcal{D}} \left[\|d_{\boldsymbol{\theta}'}(\boldsymbol{\epsilon}) - \boldsymbol{x}_0\|^2 + \lambda_{\text{train}}\|\mathcal{R}(\boldsymbol{x})\|^2\right],\end{aligned} \tag{5}$$

where $\lambda_{\text{train}}$ balances generative fidelity and physical constraint satisfaction. Unlike prior work (Bastek et al., 2025; Huang et al., 2024a), the residual is evaluated on final student outputs rather than posterior-mean surrogates $\mathbb{E}[\boldsymbol{x}_0|\boldsymbol{x}_t]$, which makes the physical objective directly aligned with sample-level constraint satisfaction (see Table 4). Training is repeated until convergence (Algorithm 1). This noise-to-sample distillation is often difficult to learn due to the high curvature of sampling trajectories, yielding noise–data pairs that are far apart in Euclidean space (Liu et al., 2023c). To reduce curvature and improve learnability, we adopt linear-flow distillation (Lipman et al., 2023; Liu et al., 2023b), and we further evaluate Distribution Matching Distillation (DMD) (Yin et al., 2024), Rectified Flow, and Consistency

*Table 1.* Generative metrics on various PDE problems. PDE error is the MSE of the evaluated physics residual error. The best results are in **bold** and the second-best results are underlined.

| Dataset | Metric | **PIDDM-1** | **PIDDM-ref** | ECI | DiffusionPDE | D-Flow | PIDM | Teacher |
|---|---|---|---|---|---|---|---|---|
| Darcy | MMSE ($\times 10^{-2}$) | 0.112 | **0.037** | 0.153 | 0.419 | 0.129 | 0.515 | 0.108 |
| | SMSE ($\times 10^{-2}$) | 0.082 | **0.002** | 0.103 | 0.163 | 0.085 | 0.368 | 0.069 |
| | PDE Error ($\times 10^{-4}$) | 0.226 | **0.148** | 1.582 | 1.071 | 0.532 | 1.236 | 1.585 |
| | FPD | 0.754 | **0.385** | 0.921 | 1.437 | 0.995 | 1.983 | 0.782 |
| | NFE ($\times 10^{3}$) | **0.001** | 0.080 | 0.500 | 0.100 | 5.000 | 0.100 | 0.100 |
| Poisson | MMSE ($\times 10^{-2}$) | 0.162 | **0.113** | 0.183 | 0.861 | 0.172 | 0.948 | 0.150 |
| | SMSE ($\times 10^{-2}$) | 0.326 | **0.274** | 0.291 | 0.483 | 0.475 | 0.701 | 0.353 |
| | PDE Error ($\times 10^{-9}$) | 0.073 | **0.050** | 2.420 | 1.270 | 0.831 | 1.593 | 2.443 |
| | FPD | 1.281 | **0.659** | 1.532 | 1.835 | 1.677 | 2.358 | 1.342 |
| | NFE ($\times 10^{3}$) | **0.001** | 0.080 | 0.500 | 0.100 | 5.000 | 0.100 | 0.100 |
| Burgers | MMSE ($\times 10^{-2}$) | 0.152 | **0.012** | 0.294 | 0.064 | 0.305 | 0.948 | 0.264 |
| | SMSE ($\times 10^{-2}$) | 0.133 | **0.101** | 0.105 | 0.103 | 0.207 | 0.701 | 0.114 |
| | PDE Error ($\times 10^{-3}$) | 0.466 | **0.174** | 1.572 | 1.032 | 0.730 | 1.593 | 1.334 |
| | FPD | 0.129 | **0.054** | 0.387 | 1.133 | 0.695 | 1.437 | 0.118 |
| | NFE ($\times 10^{3}$) | **0.001** | 0.080 | 0.500 | 0.100 | 5.000 | 0.100 | 0.100 |

Model (Song et al., 2023b) to strengthen coupling and distribution alignment. These choices produce consistent gains (see Table 4).

To clarify the optimization mechanism, Appendix B.3 analyzes a low-rank Gaussian target, where the low-rank direction mimics a hard equality constraint in PDE problems. The main intuition is that diffusion training and posterior-mean guidance operate in score space along a noisy trajectory. Near the clean endpoint, the score component associated with the constrained direction becomes increasingly sharp, so small approximation errors in this component can persist as visible constraint violations in generated samples. PIDDM instead moves the physics penalty to data space after the endpoint sample is formed. The student therefore does not need to resolve the sharp score field at every noise level; it is trained to suppress the violating component of its final output directly. This explains why final-sample distillation can reduce sample-level PDE residuals, while our distributional fidelity claims remain empirical rather than an end-to-end theoretical guarantee.

### 3.3. Downstream Tasks

Our method naturally supports one-step generation of physically constrained data, jointly producing both coefficient and solution fields. Beyond this intrinsic functionality, it also retains the flexibility of the teacher diffusion model, enabling various downstream tasks such as forward and inverse problem solving, and reconstruction from partial observations. Compared to the teacher model, our method achieves these capabilities with improved computational efficiency and stronger physical alignment.

**Generative Modeling.** We aim to sample physically consistent pairs $x_0 = (u, a)$ from a learned distribution that satisfies the governing PDE system. The student model supports this via efficient one-step generation: given $\epsilon \sim \mathcal{N}(0, I)$, it outputs $x_0 = d_{\theta'}(\epsilon)$, approximating a valid solution–coefficient pair. We further provide an optional refinement stage based on constraint-driven optimization (Algorithm 2), which reduces the physics residual by updating $\epsilon$ with gradient descent. This design is inspired by noise prompting methods (Ben-Hamu et al., 2024; Guo et al., 2024a) that optimize the final sample with respect to the initial noise. However, in contrast to those prior works, which backpropagate through an entire sampling trajectory and incur high cost and unstable gradients, our refinement operates in a one-step setting. While optional, it offers additional control that is useful in scientific applications requiring strict physical consistency (LeVeque & Leveque, 1992; Hansen et al., 2023; Mouli et al., 2024; Saad et al., 2022).

**Forward/Inverse Problem and Reconstruction.** PIDDM handles all downstream problems as conditional generation over the joint field $x = (u, a)$. Forward inference draws $u$ from known $a$; inverse inference recovers $a$ from observed $u$; reconstruction fills in missing entries of $(u, a)$ given a partial observation $x'$. We solve this via optimization-based inference on the latent variable $\varepsilon$, using the same student model $d_{\theta'}$ as in generation, as described in Algorithm 3. Let $x = d_{\theta'}(\varepsilon)$ denote the generated sample, and let $M$ be a binary observation mask indicating the known entries in $x'$ with respect to $x$. To ensure hard consistency with observed values (e.g., boundary conditions $\mathcal{B}$), we define a mixed sample by injecting observed entries into the gener-

ated output, following ECI-sampling (Cheng et al., 2025), and then update $\varepsilon$ using a combined objective:

$$\mathcal{L}_{\text{total}} = \|(\boldsymbol{x} - \boldsymbol{x}') \odot M\|^2 + \lambda \|\mathcal{R}(\boldsymbol{x}_{\text{mix}})\|^2, \quad (6)$$
$$\boldsymbol{x}_{\text{mix}} = \boldsymbol{x}' \odot M + \boldsymbol{x} \odot (1 - M).$$

Interestingly, we also find that applying this masking not only enhances hard constraints on $\mathcal{B}$, but also improves satisfaction of $\mathcal{F}$, as demonstrated in our ablation study in Table 4. Classical inverse solvers (Li et al., 2020; 2024; Lu et al., 2019; Raissi et al., 2019) learn a deterministic map $\boldsymbol{u} \mapsto \boldsymbol{a}$ and therefore require full observations of $\boldsymbol{a}$ to evaluate $\mathcal{F}[\boldsymbol{u}, \boldsymbol{a}] = 0$, a condition rarely met in practice. DiffusionPDE (Huang et al., 2024a) relaxes this by sampling missing variables, but enforces physics on the posterior mean, i.e. $\mathcal{F}\big[\mathbb{E}[\boldsymbol{x}_0|\boldsymbol{x}_t]\big]$, and thus suffers from the Jensen's Gap. Our method avoids this inconsistency by imposing constraints directly on the final sample $\mathcal{F}[\boldsymbol{x}_0]$, yielding more reliable and physically consistent inverse solutions.

# 4. Experiments

**Experiment Setup.** We consider three widely used PDE benchmarks in the main text: Darcy flow, Poisson equation, and Burgers' equation. These datasets are readily accessible from FNO (Li et al., 2020) and DiffusionPDE (Huang et al., 2024a). We also provide results on other benchmarks in Appendix E. We consider ECI (Cheng et al., 2025), DiffusionPDE (Huang et al., 2024a), D-Flow (Cheng et al., 2025; Ben-Hamu et al., 2024), PIDM (Bastek et al., 2025), and vanilla teacher diffusion models as baseline methods, with detailed implementations in Appendix D.4. We follow ECI-sampling (Cheng et al., 2025) to use FNO as both the teacher diffusion model and the student distillation model. We provide the full specification of our experiment setup in Appendix D.

To quantitatively evaluate generative performance, we report MMSE, SMSE, FPD, and PDE error following prior work (Cheng et al., 2025; Kerrigan et al., 2023; Bastek et al., 2025; Jacobsen et al., 2024): MMSE measures the mean squared error of the sample mean; SMSE evaluates the error of the sample standard deviation; FPD evaluates the Frechet distance between hidden representations extracted by the pre-trained PDE foundation model; PDE error quantifies the violation of physical constraints using the physics residual error $|\mathcal{R}(\boldsymbol{x})|^2$. The number of function evaluations (NFE) reflects computational cost during inference. For downstream tasks, we further report MSE on solution fields, coefficient fields, or both, depending on the problem setting, reflecting the accuracy of PDE solving.

## 4.1. Empirical Evaluations

PIDDM samples the joint field $(\boldsymbol{u}, \boldsymbol{a})$, enabling forward $(\boldsymbol{u}|\boldsymbol{a})$, inverse $(\boldsymbol{a}|\boldsymbol{u})$, and reconstruction (partial $\boldsymbol{u}, \boldsymbol{a}$) tasks

*Table 2.* Evaluation on various downstream tasks on Darcy datasets. PDE error is the MSE of the evaluated physics residual error. The units of MSE, PDE error, and NFE are $\times 10^{-1}$, $\times 10^{-4}$, and $\times 10^3$, respectively. The best results are in **bold**.

| Task | Metric | **PIDDM** | ECI | DiffusionPDE | D-Flow | PIDM |
|------|--------|-----------|-----|--------------|--------|------|
| Forward | MSE | **0.316** | 0.776 | 0.691 | 0.539 | 0.380 |
| | PDE Error | **0.145** | 1.573 | 1.576 | 0.584 | 1.248 |
| | NFE | **0.080** | 0.500 | 0.100 | 5.000 | 0.100 |
| Inverse | MSE | **0.236** | 0.545 | 0.456 | 0.428 | 0.468 |
| | PDE Error | **0.126** | 1.505 | 1.402 | 0.438 | 1.113 |
| | NFE | **0.080** | 0.500 | 0.100 | 5.000 | 0.100 |
| Reconstruct | Coef MSE | **0.128** | 0.395 | 0.240 | 0.158 | 0.179 |
| | Sol MSE | **0.102** | 0.219 | 0.143 | 0.125 | 0.147 |
| | PDE Error | **0.143** | 1.205 | 1.239 | 0.605 | 1.240 |
| | NFE | **0.080** | 0.500 | 0.100 | 5.000 | 0.100 |

(Sec. 3.3). DiffusionPDE (Huang et al., 2024a) reports only reconstruction MSE, while ECI-sampling (Cheng et al., 2025) and PIDM (Bastek et al., 2025) cover at most one task, limited to either unconditional generation or forward solving. For a fair comparison, we evaluate all methods on all three tasks, providing a unified view of generative quality and physical fidelity.

**Generative Tasks.** We first evaluate the generative performance of our method across three representative PDE systems: Darcy, Poisson, and Burgers' equation. As shown in Table 1, our one-step model (*PIDDM-1*) achieves competitive MMSE and SMSE scores while maintaining extremely low computational cost (1 NFE). Notably, *PIDDM-1 already surpasses prior methods that incorporate physical constraints during training or sampling*, such as PIDM, DiffusionPDE, and ECI-sampling, which impose constraints on posterior-mean surrogates and only exhibit marginal improvements over vanilla diffusion baselines. *Our optional refinement stage (PIDDM-ref) further reduces both statistical errors and physical PDE residuals, outperforming all baselines.* Meanwhile, ECI, which only enforces hard constraints on boundary conditions, achieves moderate improvements but remains less effective on field-level physical consistency. Although D-Flow enforces physical constraints through final-sample trajectory optimization, it requires thousands of NFEs and often suffers from gradient instability.

**Compute-Matched Comparison.** Table 3 further compares inference cost and quality on Darcy generation under the same hardware. PIDDM-1 uses one forward evaluation and no backward pass, giving millisecond-level generation while improving PDE loss and distributional metrics over posterior-mean-guided baselines. With the same 80-step online budget as DiffusionPDE, PIDDM-ref uses final-sample refinement to obtain lower PDE loss, FPD, MMSE, and SMSE. D-Flow optimizes the final sample directly but requires 2000 forward and backward passes, making it sub-

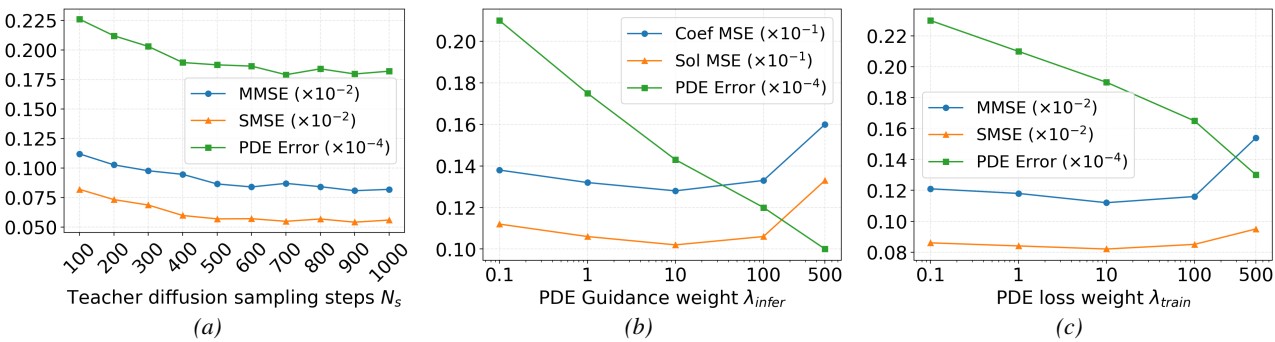

*Figure 3.* Ablation studies on the effect of several factors on the performance of PIDDM on the Darcy dataset. Panels (a), (b), and (c) show the effects of $N_s$, $\lambda_{\text{train}}$, and $\lambda_{\text{infer}}$, respectively.

*Table 3.* Compute-matched Darcy generation. Best or tied-best entries are in **bold**.

| Method | Time | NFE | BWD | PDE | FPD | MMSE | SMSE |
|---|---|---|---|---|---|---|---|
| PIDDM-1 | **0.004** | **1** | **0** | 0.226 | 0.754 | 0.112 | 0.082 |
| PIDDM-ref | 1.170 | 80 | 80 | **0.148** | **0.385** | **0.037** | **0.002** |
| DiffPDE | 1.250 | 80 | 80 | 1.129 | 1.452 | 0.431 | 0.173 |
| PIDM | 1.281 | 300 | 0 | 1.274 | 2.018 | 0.522 | 0.374 |
| D-Flow | 19.302 | 2000 | 2000 | 0.885 | 1.252 | 0.254 | 0.209 |
| ECI | 1.294 | 300 | 0 | 1.106 | 0.837 | 0.138 | 0.112 |

Time is wall-clock seconds per sample; PDE, MMSE, and SMSE are scaled as in Table 1.

stantially slower despite weaker statistical metrics. Here, NFE denotes the number of forward function evaluations, and BWD denotes the number of inference-time backward passes.

**Forward/Inverse Solving and Reconstruction.** We further demonstrate the versatility of our method in forward and inverse problem solving on the Darcy dataset. Since the original PIDM (Bastek et al., 2025) implementation addresses only unconditional generation, we pair it with Diffusion Posterior Sampling (DPS) (Chung et al., 2023) to extend it to downstream tasks (forward, inverse, and reconstruction). Following the test protocol of D-Flow, we apply inference-time optimization over the initial noise to match given observations while satisfying physical laws. As shown in Table 2, *our method (**PIDDM**) achieves the best results across all metrics*, including MSE and PDE error, while being significantly more efficient than D-Flow, which requires 5000 function evaluations. Compared to ECI and DiffusionPDE, our method yields lower residuals and better predictive accuracy, reflecting its superior handling of physical and observational constraints jointly.

### 4.2. Ablation Studies

To better understand the effect of key design choices in PIDDM, we perform ablations on five factors: teacher sampling steps $N_s$; distillation weight $\lambda_{\text{train}}$; inference weight $\lambda_{\text{infer}}$; diffusion schedule (VP, sub-VP, linear); and advanced

distillation variants (Rectified Flow, DMD, Consistency Model). Figure 3 presents three key ablation studies on the Darcy dataset. Panel (a) shows that increasing the teacher model's sampling steps $N_s$ consistently improves both generative quality and physical alignment, as reflected by lower MMSE, SMSE, and PDE residuals of the distilled student, highlighting the importance of high-fidelity supervision. Panels (b) and (c) examine the impact of the PDE loss weight during distillation and inference, respectively. We observe that across a wide range of weights, all metrics achieve strong performance. In particular, compared with baselines in Table 2, the PDE error is reduced by an order of magnitude. This improvement is consistent with the design goal of enforcing PDE constraints on final generated samples rather than posterior-mean surrogates.

We also explore whether more sophisticated distillation strategies can improve the quality of the student model. As shown in Table 4, advanced techniques such as Rectified Flow (RF-1, RF-2), Distribution Matching Distillation (DMD), and Consistency Model yield better MMSE and SMSE than our raw method, while maintaining competitive PDE residuals. This indicates that tighter coupling between noise and data trajectories during distillation facilitates noise-data learning. In addition, we analyze the effect of imposing hard constraints during downstream inference. Following the strategy inspired by ECI-sampling, we directly replace the masked entries in the generated sample with observed values before computing the PDE residual. This ensures that the known information is preserved when evaluating physical consistency. As shown by the $-$HC variant in Table 4, removing this hard-constraint replacement significantly degrades the PDE residual errors across all tasks. We also validate our design using other linear diffusion processes in Table 4 (i.e., VP and sub-VP).

Appendix G provides additional diagnostics for the main design choices. Table 6 separates one-step distillation from final-sample PDE supervision, showing that distillation mainly gives speed while final-sample supervision drives physical consistency; Table 13 confirms the same pattern

*Table 4.* Evaluation on various downstream tasks on Darcy datasets under different PIDDM settings. PIDDM denotes the raw method, +RF-1 and +RF-2 denote one and two rounds of reflowing (Liu et al., 2023c), +DMD denotes distribution matching distillation (Yin et al., 2024), and +CM denotes the consistency model (Song et al., 2023b). −HC refers to the ablation without the hard-constraint replacement in Eq. 6 during PIDDM inference. VP and sub-VP refer to ablations on VP and sub-VP diffusion processes. PDE error is the MSE of the evaluated physics residual error. The best results are in **bold**.

| Task | Metric | PIDDM | +RF-1 | +RF-2 | +DMD | +CM | −HC | VP | sub-VP |
|---|---|---|---|---|---|---|---|---|---|
| Forward | MSE ($\times 10^{-1}$) | 0.316 | 0.278 | **0.127** | 0.255 | 0.283 | 0.705 | 0.398 | 0.372 |
| | PDE Error ($\times 10^{-4}$) | 0.145 | 0.129 | 0.098 | 0.134 | **0.083** | 0.354 | 0.154 | 0.157 |
| Inverse | MSE ($\times 10^{-1}$) | 0.236 | 0.195 | **0.136** | 0.188 | 0.182 | 0.503 | 0.284 | 0.271 |
| | PDE Error ($\times 10^{-4}$) | 0.115 | 0.126 | **0.079** | 0.121 | 0.109 | 0.321 | 0.143 | 0.139 |
| Reconstruct | Coef MSE ($\times 10^{-1}$) | 0.128 | 0.107 | 0.091 | 0.095 | **0.085** | 0.294 | 0.133 | 0.138 |
| | Sol MSE ($\times 10^{-1}$) | 0.102 | 0.084 | **0.063** | 0.073 | 0.072 | 0.239 | 0.127 | 0.119 |
| | PDE Error ($\times 10^{-4}$) | 0.143 | 0.118 | 0.085 | 0.104 | **0.083** | 0.464 | 0.159 | 0.158 |

across RF, DMD, and consistency-model backbones. Table 7 further shows that PIDDM preserves sample diversity better than vanilla distillation. We also test robustness to teacher quality and architecture in Tables 8 and 9, and report dynamic weighting and latent-optimization stability in Tables 10–12.

## 5. Conclusion and Limitation

We propose **PIDDM**, a lightweight post-hoc distillation framework for physics-constrained diffusion models. Unlike methods that impose PDE constraints on the posterior mean, leading to a Jensen's Gap and a trade-off between quality and constraint satisfaction, PIDDM enforces constraints directly on the final generated sample. This design improves sample-level physical consistency while empirically preserving distributional fidelity. We provide empirical illustrations of the Jensen's Gap in diffusion training and sampling. Experiments show PIDDM improves physical and distributional fidelity across forward, inverse, and partial reconstruction tasks, with robustness to hyperparameter choice. The distilled student model also enables efficient one-step physics simulation for repeated-query and latency-sensitive scientific workflows such as uncertainty quantification, online data assimilation, virtual sensing, and design optimization.

**Limitations.** PIDDM requires a well-trained teacher model and a reliable PDE residual operator, which can be challenging to obtain. The one-step student's performance may degrade if the teacher is poorly calibrated or lacks diversity. Although performance is robust across a wide range of PDE loss weights, some hyperparameter tuning is still needed to obtain optimal results. Addressing these limitations remains important future work.

## Acknowledgements

We sincerely thank the reviewers and the area chair for their careful reading, constructive feedback, and thoughtful suggestions, which helped us improve the clarity and presentation of this work. D. Zou and Y. Zhang are supported in part by NSFC 62306252, Hong Kong ECS award 27309624, Guangdong NSF 2024A1515012444, and the central fund from HKU. P. Wang is supported in part by the University of Macau SRG2025-00043-FST and UMDF-TISF-I/2026/013/FST, and in part by the Macau Science and Technology Development Fund (FDCT) 0091/2025/ITP2.

## Impact Statement

This work makes physics-constrained diffusion models practical by enforcing PDE constraints post hoc, sidestepping the posterior-mean mismatch in the student objective and enabling fast single-step generation. By improving physical fidelity and reducing compute and tuning burden, PIDDM can accelerate simulation, inverse design, and data completion workflows in engineering and scientific domains.

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

# A. Related Work

## A.1. Diffusion Models

Diffusion models (Song et al., 2020; Ho et al., 2020; Karras et al., 2022) learn a score function, $\nabla \log p(\boldsymbol{x}_t)$, to reverse a predefined diffusion process, typically of the form $\boldsymbol{x}_t = \boldsymbol{x}_0 + \sigma_t \boldsymbol{\varepsilon}$. A key characteristic of diffusion models is that sampling requires iteratively reversing this process over a sequence of timesteps. This iterative nature presents a challenge for controlled generation: to guide the sampling trajectory effectively, we often need to first estimate the current denoised target $x_0$ in order to determine the correct guidance direction. In other words, *to decide how to get there, we must first understand where we are*. However, obtaining this information through full iterative sampling is computationally expensive and often impractical in an optimization regime.

A practical workaround is to leverage an implicit one-step data estimate provided by diffusion models via Tweedie's formula (Efron, 2011), which requires only a single network forward pass:

$$\hat{\boldsymbol{x}}_0 \approx \mathbb{E}[\boldsymbol{x}_0|\boldsymbol{x}_t] = \boldsymbol{x}_t + \sigma_t^2 \nabla \log p(\boldsymbol{x}_t),$$

where $\hat{x}_0$ denotes a one-step denoised estimate of the clean sample from $\boldsymbol{x}_t$. This approximation improves as $t \to 0$. Although this posterior mean $\mathbb{E}[\boldsymbol{x}_0 \mid \boldsymbol{x}_t]$ is not theoretically equivalent to the final sample obtained after full denoising, in practice, this estimate serves as a useful proxy for the underlying data and enables approximate guidance for controlled generation, without the need to complete the entire sampling trajectory.

## A.2. Constrained Generation for PDE Systems

Diffusion models have demonstrated strong potential for physical-constraint applications due to their generative nature. This generative capability naturally supports the basic task of simulating physical data and also extends to downstream applications such as reconstruction from partial observations and solving both forward and inverse problems. However, many scientific tasks require strict adherence to physical laws, often expressed as PDE constraints on the data. These constraints, applied at the sample level $\boldsymbol{x}$, are not easily enforced within diffusion models, which are trained to model the data distribution $p(\boldsymbol{x})$. To address this, prior works have proposed three main strategies for incorporating physical constraints into diffusion models.

**Training-time Loss Injection.** PG-Diffusion (Shu et al., 2023) employs Classifier-Free Guidance (CFG), where a conditional diffusion model is trained using the PDE residual error as a conditioning input. However, CFG is well known to suffer from theoretical inconsistencies—specifically, the interpolated conditional score function does not match the true conditional score—which limits its suitability for enforcing precise physical constraints. To avoid this issue, PIDM (Bastek et al., 2025) introduces a loss term based on the residual evaluated at the posterior mean, $\mathbb{E}[\boldsymbol{x}_0 \mid \boldsymbol{x}_t]$. While this approach avoids the theoretical pitfalls of CFG, the constraint is still not imposed on the actual sample $\boldsymbol{x}_0$, inheriting the Jensen-gap issue originally discussed in diffusion posterior sampling for inverse problems (Chung et al., 2023).

**Sampling-time Guidance.** Diffusion Posterior Sampling (DPS), used in DiffusionPDE (Huang et al., 2024a) and CoCo-Gen (Jacobsen et al., 2024), applies guidance during each sampling step by using the gradient of the PDE residual evaluated on the posterior mean $\mathbb{E}[\boldsymbol{x}_0 \mid \boldsymbol{x}_t]$. Therefore, they inherit the Jensen's Gap issue, as the guidance operates on an estimate of the final sample rather than the sample itself. Moreover, DPS assumes that the residual error follows a Gaussian distribution—a condition that may not hold in real-world PDE systems. Meanwhile, to support hard constraints, ECI-sampling (Cheng et al., 2025) directly modifies the posterior mean $\mathbb{E}[\boldsymbol{x}_0 \mid \boldsymbol{x}_t]$ using known boundary conditions.

**Noise Prompting.** Another stream of research—often called *noise prompting* or *golden-noise optimization*—directly tunes the *initial* noise so that the resulting sample satisfies a target constraint (Ben-Hamu et al., 2024; Guo et al., 2024a; Zhou et al., 2024b; Wang et al., 2024; Mao et al., 2024; Chen et al., 2024). In the physics domain, this idea is used to minimize the true PDE residual $R(\boldsymbol{x})$ evaluated on the *final* sample, rather than the surrogate residual $R(\mathbb{E}[\boldsymbol{x}_0 \mid \boldsymbol{x}_t])$. Because the constraint is imposed on the actual output, noise prompting avoids posterior-mean surrogate constraints and therefore serves as a strong baseline in ECI-sampling (Cheng et al., 2025) and PIDM (Bastek et al., 2025). The main drawback is efficiency: optimizing the noise requires backpropagating through the entire sampling trajectory, which is computationally expensive and prone to gradient instability.

Recently, diffusion-based techniques for solving image inverse problems have demonstrated competitive performance (Zheng et al., 2025). However, a common limitation is that they rely on the posterior mean, i.e., $\mathbb{E}[\boldsymbol{x}_0 \mid \boldsymbol{x}_t]$, as a surrogate for the true posterior. For example, DDRM (Kawar et al., 2022) and DDNM (Wang et al., 2022) exploit singular value

decomposition (SVD) and pseudo-inverse operations to fill in the missing components of $\mathbb{E}[\boldsymbol{x}_0 \mid \boldsymbol{x}_t]$ during sampling, which is conceptually similar to ECI-sampling. Likewise, methods such as DiffPIR (Zhu et al., 2023) and DAPS (Zhang et al., 2025) optimize or run Langevin MCMC updates on the posterior mean in order to enforce observation consistency. Another line of work approximates the likelihood $p(\boldsymbol{y} \mid \boldsymbol{x}_t)$. For instance, DPS (Chung et al., 2023) treats $p(\boldsymbol{x}_0 \mid \boldsymbol{x}_t)$ as a point mass centered at $\mathbb{E}[\boldsymbol{x}_0 \mid \boldsymbol{x}_t]$, while LGD (Song et al., 2023a) and DPG (Tang et al., 2024) use a Gaussian distribution with mean $\mathbb{E}[\boldsymbol{x}_0 \mid \boldsymbol{x}_t]$ for approximation. To reduce this approximation error, some methods trade off computational cost: Monte Carlo–based approaches (Dou & Song, 2024; Cardoso et al., 2024; Huang et al., 2024b) and variational inference–based approaches (Wu et al., 2024) avoid the direct mean approximation, but they either require simulating a large number of samples or solving intermediate optimization problems during sampling, both of which are computationally expensive. Finally, there are methods that explicitly aim to reduce the Jensen's Gap by modifying the sampling dynamics. Examples include mid-point schemes (Moufad et al., 2024) and user-defined intermediate potentials (Dong et al., 2024). While these approaches can shorten the gap, they still rely on $\mathbb{E}[\boldsymbol{x}_0 \mid \boldsymbol{x}_t]$ for posterior estimation, and moreover, they often incur high computational cost due to additional variational inference or Langevin MCMC steps.

### A.3. Distillation of Diffusion Models

Sampling in diffusion models involves integrating through a reverse diffusion process, which is computationally expensive. Even with the aid of high-order ODE solvers (Maoutsa et al., 2020; Song et al., 2021; Lu et al., 2022a;b; Zhou et al., 2024a), parallel sampling (Shih et al., 2024), and better training schedules (Karras et al., 2022; Nichol & Dhariwal, 2021; Liu et al., 2023a;b;c), the process remains iterative and typically requires hundreds of network forward passes. To alleviate this inefficiency, distillation-based methods have been developed to enable one-step generation by leveraging the deterministic nature of samplers (e.g., DDIM), where the noise–data pairs become fixed. The most basic formulation, Knowledge Distillation (Luhman & Luhman, 2021), trains a student model to replicate the teacher's deterministic noise-to-data mapping. However, subsequent studies have shown that directly learning this raw mapping is challenging for neural networks, as the high curvature of sampling trajectories often yields noise–data pairs that are distant in Euclidean space, making the regression task ill-conditioned and hard to generalize.

To address this, recent research has proposed three complementary strategies. (1) Noise–data coupling refinement: Rectified Flow (Liu et al., 2023b) distills the sampling process into a structure approximating optimal transport, where the learned mapping corresponds to minimal-cost trajectories between noise and data. InstaFlow (Liu et al., 2023c) further demonstrates that such near-optimal-transport couplings significantly ease the learning process for student models. (2) Distribution-level distillation: Rather than matching individual noise–data pairs, DMD (Yin et al., 2024) trains the student via score-matching losses that align the overall data distributions, thereby bypassing the need to regress complex mappings directly. (3) Trajectory distillation: Instead of only supervising on initial ($\boldsymbol{x}_T$) and final ($\boldsymbol{x}_0$) states, this approach provides supervision at intermediate states $x_t$ along the ODE trajectory (Berthelot et al., 2023; Zheng et al., 2023; Song et al., 2023b; Tee et al., 2024). This decomposition allows the student model to learn the generative process in a piecewise manner, which improves stability and sample fidelity. We note that among existing approaches, Physics-Informed Distillation (PID) (Tee et al., 2024) has a name similar to our method's but differs fundamentally in both objective and methodology. Specifically, PID distills ODE trajectories from teacher models using a PINN-like strategy, whereas our method distills diffusion models for PDE-constrained generation by applying physical supervision directly to the final samples.

## B. Mixture-of-Gaussians (MoG) Dataset

To study the sampling-time behavior of constrained diffusion models, we design a synthetic 2D Mixture-of-Gaussians (MoG) dataset with analytical score functions. Each sample $x = (x_1, x_2) \in \mathbb{R}^2$ consists of a data dimension $x_1$ and a fixed latent code $x_2$ that serves as a hard constraint.

Specifically, we define a mixture model where $x_1$ is sampled from a Gaussian mixture conditioned on the latent code $z \in \{-1, +1\}$, and $x_2$ is deterministically set to $z$. The full distribution is:

$$x_2 = z \in \{-1, +1\}, \quad x_1 \sim \mathcal{N}(\mu_z, \sigma^2), \tag{7}$$

with $\mu_{-1} = -1$, $\mu_{+1} = +1$, and fixed standard deviation $\sigma = 0.2$. The full 2D data point is thus given by:

$$x = \begin{bmatrix} x_1 \\ x_2 \end{bmatrix}, \quad \text{with } x_1 \sim \mathcal{N}(\mu_{x_2}, \sigma^2), \quad x_2 \in \{-1, +1\}. \tag{8}$$

The resulting joint density $p(x)$ is a mixture of two Gaussians supported on parallel horizontal lines:

$$p(x) = \frac{1}{2} \mathcal{N}(x_1; -1, \sigma^2) \cdot \delta(x_2 + 1) + \frac{1}{2} \mathcal{N}(x_1; +1, \sigma^2) \cdot \delta(x_2 - 1), \tag{9}$$

where $\delta(\cdot)$ denotes the Dirac delta function. In our experiment comparing DPS in Sec. 2.3, we tune the weight of DPS guidance to be 0.035, as it gives stable performance.

## B.1. Derivation of Score Function of the MoG Dataset

MoG distributions admit analytical diffusion objectives. Specifically, consider a MoG with the form:

$$\boldsymbol{x}_0 \sim \frac{1}{K} \sum_{k=1}^{K} \mathcal{N}(\boldsymbol{\mu}_k, \sigma_k^2 \cdot \boldsymbol{I}),$$

where $K$ is the number of Gaussian components, $\boldsymbol{\mu}_k$ and $\sigma_k^2$ are the means and variances of the Gaussian components, respectively. Suppose the solution of the diffusion process follows:

$$\boldsymbol{x}_t = \alpha_t \boldsymbol{x}_0 + \sigma_t \cdot \xi \quad \text{where} \quad \xi \sim \mathcal{N}(0, \boldsymbol{I}).$$

Since $\boldsymbol{x}_0$ and $\xi$ are both sampled from Gaussian distributions, their linear combination $\boldsymbol{x}_t$ also forms a Gaussian distribution, i.e.,

$$\boldsymbol{x}_t \sim \frac{1}{K} \sum_{k=1}^{K} \mathcal{N}(\alpha_t \boldsymbol{\mu}_k, (\sigma_k^2 \alpha_t^2 + \sigma_t^2) \cdot \boldsymbol{I}).$$

Let $p_i(\boldsymbol{x}_t) = \mathcal{N}(\boldsymbol{x}_t; \alpha_t \boldsymbol{\mu}_i, (\sigma_i^2 \alpha_t^2 + \sigma_t^2)\boldsymbol{I})$. Then,

$$\nabla p_t(\boldsymbol{x}_t) = \frac{1}{K} \sum_{i=1}^{K} \nabla_{\boldsymbol{x}_t} p_i(\boldsymbol{x}_t)$$

$$= \frac{1}{K} \sum_{i=1}^{K} p_i(\boldsymbol{x}_t) \cdot \frac{-(\boldsymbol{x}_t - \alpha_t \boldsymbol{\mu}_i)}{\sigma_i^2 \alpha_t^2 + \sigma_t^2}.$$

We can also calculate the score of $\boldsymbol{x}_t$, i.e.,

$$\nabla \log p_t(\boldsymbol{x}_t) = \frac{\nabla p_t(\boldsymbol{x}_t)}{p_t(\boldsymbol{x}_t)} = \frac{\sum_{i=1}^{K} p_i(\boldsymbol{x}_t) \cdot \left( \frac{-(\boldsymbol{x}_t - \alpha_t \boldsymbol{\mu}_i)}{\sigma_i^2 \alpha_t^2 + \sigma_t^2} \right)}{\sum_{i=1}^{K} p_i(\boldsymbol{x}_t)}.$$

## B.2. Deviation of Velocity Field of Reverse ODE and DPS

Diffusion models define a forward diffusion process to perturb the data distribution $p_{\text{data}}$ to a Gaussian distribution. Formally, the diffusion process is an Itô SDE $d\boldsymbol{x}_t = f(\boldsymbol{x}_t) + g(t)d\mathbf{w}$, where $d\mathbf{w}$ is the Brownian motion and $t$ flows forward from 0 to $T$. The solution of this diffusion process gives a transition distribution $p_t(\boldsymbol{x}_t|\boldsymbol{x}_0) = \mathcal{N}(\boldsymbol{x}_t|\alpha_t \boldsymbol{x}_0, \sigma_t^2 \boldsymbol{I})$, where $\alpha_t = \exp\left(\int_0^t f(s)ds\right)$ and $\sigma_t^2 = 1 - \exp\left(-\int_0^t g(s)^2 ds\right)$. For the linear interpolation used in our experiments, $\alpha_t = 1 - t$ and $\sigma_t = t$. To sample from the diffusion model, a typical approach is to apply a reverse-time SDE that reverses the diffusion process (Anderson, 1982):

$$d\boldsymbol{x}_t = \left(f(\boldsymbol{x}_t) - g(t)^2 \nabla_{\boldsymbol{x}_t} \log p_t(\boldsymbol{x}_t)\right) dt + g(t)d\bar{\mathbf{w}},$$

where $d\bar{\mathbf{w}}$ is the Brownian motion and $t$ flows backward from $T$ to 0. For all reverse-time SDEs, there exist corresponding deterministic processes that share the same density evolution, i.e., $\{p_t(\boldsymbol{x}_t)\}_{t=0}^{T}$ (Song et al., 2020). Specifically, this deterministic process follows an ODE:

$$d\boldsymbol{x}_t = \left(f(\boldsymbol{x}_t) - \frac{1}{2} g(t)^2 \nabla_{\boldsymbol{x}_t} \log p_t(\boldsymbol{x}_t)\right) dt,$$

where $t$ flows backward from $T$ to $0$. The deterministic process defines a velocity field,

$$v_{\text{GT}}(\boldsymbol{x}, t) = [f(\boldsymbol{x}_t) - \frac{1}{2}g(t)^2 \nabla_{\boldsymbol{x}_t} \log p_t(\boldsymbol{x}_t)].$$

Here, we also define the velocity field by $v(\boldsymbol{x}_t, t) = f(\boldsymbol{x}_t) - \frac{1}{2}g(t)^2 \nabla_{\boldsymbol{x}_t} \log p_t(\boldsymbol{x}_t)$.

The posterior mean can be estimated from the score by:

$$\mathbb{E}[\boldsymbol{x}_0 | \boldsymbol{x}_t] = \frac{\boldsymbol{x}_t + \sigma_t^2 \nabla \log p_t(\boldsymbol{x}_t)}{\alpha_t}.$$

The posterior mean can also be estimated from the velocity field by:

$$\mathbb{E}[\boldsymbol{x}_0 | \boldsymbol{x}_t] = \frac{\dot{\sigma}_t \boldsymbol{x}_t - \sigma_t v(\boldsymbol{x}_t, t)}{\alpha_t \dot{\sigma}_t - \sigma_t \dot{\alpha}_t}.$$

### B.3. Theoretical Explanation of the Effectiveness of Distillation

In this subsection, we provide a theoretical explanation for why the post-hoc distillation stage improves sampling efficiency compared to both standard diffusion models and models with low-rank guidance. Here, we consider a target distribution $\boldsymbol{x}_0 \sim \mathcal{N}(\boldsymbol{0}, \boldsymbol{\Lambda})$, where $\boldsymbol{\Lambda} = \text{diag}(\lambda_1, \ldots, \lambda_{d-1}, 0)$ is diagonal without loss of generality and $1 = \lambda_1 \geq \cdots \geq \lambda_{d-1} > 0$. The zero eigenvalue $\lambda_d = 0$ is introduced to simulate the hard equality constraint on the data, and the corresponding eigendirection $\boldsymbol{e}_d$ is referred to as the constraint direction.

We consider the variance-preserving SDE with the following forward process:

$$d\boldsymbol{x}_t = -\frac{1}{2}\beta \boldsymbol{x}_t dt + \sqrt{\beta} d\boldsymbol{w}_t, \quad 0 \leq t \leq T,$$

where $\beta > 0$ is a noise schedule and $\boldsymbol{w}_t$ is a standard Wiener process (Song et al., 2020). This implies

$$\boldsymbol{x}_t = \alpha_t \boldsymbol{x}_0 + \sigma_t \boldsymbol{\epsilon}, \text{ where } \alpha_t = \exp\left(-\frac{1}{2}\int_0^t \beta dy\right), \ \sigma_t = \sqrt{1 - \alpha_t^2}, \text{ and } \boldsymbol{\epsilon} \sim \mathcal{N}(\boldsymbol{0}, \boldsymbol{I}).$$

One can easily verify that the score function is

$$\nabla \log p_t(\boldsymbol{x}) = -\left(\alpha_t^2 \boldsymbol{\Lambda} + \sigma_t^2 \boldsymbol{I}\right)^{-1} \boldsymbol{x}.$$

For ease of exposition, let $\boldsymbol{\theta}^* := -\text{diag}\left(\alpha_t^2 \boldsymbol{\Lambda} + \sigma_t^2 \boldsymbol{I}\right)^{-1}$ denote the diagonal entries and $\nabla \log p_t(\boldsymbol{x}) = \text{diag}(\boldsymbol{\theta}^*)\boldsymbol{x}$. In particular, along the constrained direction $\boldsymbol{e}_d$, we have $\theta_d^*(t) = -\sigma_t^{-2}$, whose magnitude diverges as $\sigma_t^2 \to 0$.

**Score network.** To learn the score function at each time $t$, we parameterize the score network as $s_{\boldsymbol{\theta}}(\boldsymbol{x}, t) = \text{diag}(\boldsymbol{\theta})\boldsymbol{x}$ and consider the denoising score matching loss:

$$\mathbb{E}_{\boldsymbol{x}_0, \boldsymbol{\epsilon} \sim \mathcal{N}(\boldsymbol{0}, \boldsymbol{I})}\left[\|s_{\boldsymbol{\theta}}(\boldsymbol{x}_t, t) - \nabla \log p_t(\boldsymbol{x}_t)\|^2\right].$$

Using the above loss, we learn the score function $\hat{s}_{\boldsymbol{\theta}}$ from training samples $\{\boldsymbol{x}^{(i)}\}_{i=1}^M \sim \mathcal{N}(\boldsymbol{0}, \boldsymbol{\Lambda})$ by solving the following problem:

$$\begin{aligned}
\mathcal{L}_t(\boldsymbol{\theta}) &:= \frac{1}{M}\sum_{i=1}^M \mathbb{E}_{\boldsymbol{\epsilon} \sim \mathcal{N}(\boldsymbol{0}, \boldsymbol{I})}\left[\left\|s_{\boldsymbol{\theta}}(\alpha_t \boldsymbol{x}_0^{(i)} + \sigma_t \boldsymbol{\epsilon}, t) - \nabla \log p(\boldsymbol{x}_t \mid \boldsymbol{x}_0^{(i)})\right\|^2\right] \\
&= \frac{1}{M}\sum_{i=1}^M \mathbb{E}_{\boldsymbol{\epsilon} \sim \mathcal{N}(\boldsymbol{0}, \boldsymbol{I})}\left[\left\|\text{diag}(\boldsymbol{\theta})(\alpha_t \boldsymbol{x}_0^{(i)} + \sigma_t \boldsymbol{\epsilon}) + \frac{1}{\sigma_t}\boldsymbol{\epsilon}\right\|^2\right] \\
&= \frac{1}{M}\sum_{i=1}^M \mathbb{E}_{\boldsymbol{\epsilon} \sim \mathcal{N}(\boldsymbol{0}, \boldsymbol{I})}\left[\left\|\alpha_t \text{diag}(\boldsymbol{\theta})\boldsymbol{x}_0^{(i)} + \left(\sigma_t \text{diag}(\boldsymbol{\theta}) + \frac{1}{\sigma_t}\boldsymbol{I}\right)\boldsymbol{\epsilon}\right\|^2\right] \\
&= \frac{\alpha_t^2}{M}\sum_{i=1}^M \left\|\text{diag}(\boldsymbol{\theta})\boldsymbol{x}_0^{(i)}\right\|^2 + \left\|\sigma_t \text{diag}(\boldsymbol{\theta}) + \frac{1}{\sigma_t}\boldsymbol{I}\right\|_F^2.
\end{aligned} \tag{10}$$

Moreover, the effective score function of *diffusion models with guidance* (Guo et al., 2024b) is

$$s_{\boldsymbol{\theta}}^{\mathrm{g}} = s_{\boldsymbol{\theta}}(\boldsymbol{x}_t, t) - \gamma \left\langle \frac{\boldsymbol{x}_t - \sigma_t s_{\boldsymbol{\theta}}(\boldsymbol{x}_t, t)}{\alpha_t}, \boldsymbol{e}_d \right\rangle \boldsymbol{e}_d,$$

where the guidance is designed by enforcing the predicted clean data $\hat{\boldsymbol{x}}_{0|t} = \frac{\boldsymbol{x}_t - \sigma_t s_{\boldsymbol{\theta}}(\boldsymbol{x}_t, t)}{\alpha_t}$ to have a small constraint-satisfaction loss $(\hat{\boldsymbol{x}}_{0|t}^{\top} \boldsymbol{e}_d)^2$, and $\gamma$ is the guidance strength. Based on $s_{\boldsymbol{\theta}}(\boldsymbol{x}, t) = \mathrm{diag}(\boldsymbol{\theta})\boldsymbol{x}$, one can express the guided score as $s_{\boldsymbol{\theta}}^{\mathrm{g}} = \mathrm{diag}(\boldsymbol{\theta}^g)\boldsymbol{x}$.

Then, let $\hat{\boldsymbol{x}}_0(\boldsymbol{\epsilon})$ be the data generated from the input random variable $\boldsymbol{\epsilon} \sim \mathcal{N}(0, \boldsymbol{I})$ through the ODE solver using the trained score function $s_{\boldsymbol{\theta}}$. The student network $\boldsymbol{v}_{\boldsymbol{w}}(\boldsymbol{\epsilon}) = \mathrm{diag}(\boldsymbol{w})\boldsymbol{\epsilon}$ is trained according to (5), i.e.,

$$\mathcal{L}_{\mathrm{distill}}(\boldsymbol{w}) = \mathbb{E}_{\boldsymbol{\epsilon}}[\|\boldsymbol{v}_{\boldsymbol{w}}(\boldsymbol{\epsilon}) - \hat{\boldsymbol{x}}_0(\boldsymbol{\epsilon})\|_2^2] + \lambda(\boldsymbol{v}_{\boldsymbol{w}}(\boldsymbol{\epsilon})^{\top} \boldsymbol{e}_d)^2.$$

**Constrained direction may be imperfectly learned via vanilla diffusion models and diffusion models with guidance.** Based on the above setup, we can analyze the learning ability of these methods in terms of the hard constraint on the final coordinate. In particular, we will show that even in the simple linearized setting, learning the constrained coordinate $\theta_d^*(t)$ will be slow at small noise levels, i.e., $t \ll 1$.

**Theorem B.1.** *Suppose that the number of training samples $M$ is sufficiently large and we apply gradient descent to solve the empirical loss in Problem (10), starting from $\boldsymbol{\theta}^0 = \mathbf{1}$ and using step size $\eta \sim \Theta(1)$ for $\ell$ iterations. Then, with high probability, for the constrained coordinate $d$, the iterates satisfy*

$$\theta_d^{\ell} - \theta_d^*(t) = \left(1 - 2\eta\sigma_t^2\right)^{\ell} \left(\theta_d^0 - \theta_d^*(t)\right), \qquad \ell = 0, 1, 2, \ldots,$$

*and consequently, for any regime with $\ell\sigma_t^2 \ll 1$,*

$$\frac{|\theta_d^{\ell} - \theta_d^*(t)|}{|\theta_d^*(t)|} = 1 - \mathcal{O}(\ell\sigma_t^2),$$

*i.e., the learned score coefficient in the constrained direction remains a* constant-factor *away from the optimum unless $\ell = \Omega(\sigma_t^{-2})$. For diffusion models with guidance, we have*

$$\theta_d^{\ell,\mathrm{g}} = \left(1 + \frac{\gamma\sigma_t}{\alpha_t}\right) \theta_d^{\ell} - \frac{\gamma}{\alpha_t}.$$

*Proof.* At time $t$, the considered denoising score matching loss (10) over the training samples $\{\boldsymbol{x}^{(i)}\}_{i=1}^{M} \sim \mathcal{N}(\mathbf{0}, \boldsymbol{\Sigma})$ is as follows:

$$\mathcal{L}_t(\boldsymbol{\theta}) = \frac{\alpha_t^2}{M} \sum_{i=1}^{M} \left\| \mathrm{diag}(\boldsymbol{\theta})\boldsymbol{x}^{(i)} \right\|^2 + \left\| \sigma_t \mathrm{diag}(\boldsymbol{\theta}) + \frac{1}{\sigma_t}\boldsymbol{I} \right\|_F^2.$$

For ease of exposition, let

$$\hat{\lambda}_j := \frac{1}{M} \sum_{i=1}^{M} \left(x_j^{(i)}\right)^2. \tag{11}$$

Note that since $\boldsymbol{\Sigma} = \mathrm{diag}(\lambda_1, \ldots, \lambda_{d-1}, 0)$ with $1 = \lambda_1 \geq \cdots \geq \lambda_{d-1} > 0$, it holds with high probability that $\hat{\lambda}_j = (1 - o(1))\lambda_j$ for all $j = 1, \ldots, d-1$ and $\hat{\lambda}_d = 0$ almost surely. Then, we have

$$\mathcal{L}_t(\boldsymbol{\theta}) = \sum_{j=1}^{d} \left( \alpha_t^2 \hat{\lambda}_j \theta_j^2 + \left( \sigma_t \theta_j + \frac{1}{\sigma_t} \right)^2 \right).$$

With initialization $\boldsymbol{\theta}^0 = \mathbf{1}_d$, the gradient descent update is

$$\boldsymbol{\theta}^{\ell+1} = \boldsymbol{\theta}^{\ell} - \eta \nabla \mathcal{L}_t(\boldsymbol{\theta}^{\ell}), \quad \ell = 0, 1, 2, \ldots.$$

This is equivalent to

$$\theta_j^{\ell+1} = \theta_j^\ell - 2\eta \left( \left( \alpha_t^2 \hat{\lambda}_j + \sigma_t^2 \right) \theta_j^\ell + 1 \right)$$
$$= \left( 1 - 2\eta \left( \alpha_t^2 \hat{\lambda}_j + \sigma_t^2 \right) \right) \theta_j^\ell - 2\eta, \ \forall j = 1, \dots, d.$$

Obviously, the optimal solution is

$$\theta_j^* = -\frac{1}{\alpha_t^2 \hat{\lambda}_j + \sigma_t^2}, \ \forall j = 1, \dots, d.$$

Then, we have

$$\theta_j^{\ell+1} - \theta_j^* = \left( 1 - 2\eta \left( \alpha_t^2 \hat{\lambda}_j + \sigma_t^2 \right) \right) \left( \theta_j^\ell - \theta_j^* \right).$$

To guarantee monotonically decreasing for $j = 1$, we have

$$\eta < \frac{1}{2 \left( \alpha_t^2 \hat{\lambda}_1 + \sigma_t^2 \right)} \approx \frac{1}{2},$$

where the last inequality follows from $\hat{\lambda}_1 \approx 1$. Then, we set $\eta = 1/4$. Next, we have

$$\theta_d^{\ell+1} - \theta_d^* = \left( 1 - 2\eta \sigma_t^2 \right) \left( \theta_d^\ell - \theta_d^* \right) = \left( 1 - \frac{\sigma_t^2}{2} \right) \left( \theta_d^\ell - \theta_d^* \right).$$

This, together with $\theta_d^0 = 1$, implies for all $\ell = 1, 2, \dots,$

$$\theta_d^\ell - \theta_d^* = \left( 1 - \frac{\sigma_t^2}{2} \right)^\ell \left( 1 - \theta_d^* \right).$$

Then by the approximation $(1 - 2\eta \sigma_t^2)^\ell = 1 - \mathcal{O}(\ell \sigma_t^2)$ when $\ell \sigma_t^2 \ll 1$, we can obtain

$$\frac{|\theta_d^\ell - \theta_d^*(t)|}{|\theta_d^*(t)|} = 1 - \mathcal{O}(\ell \sigma_t^2).$$

Thus, the learned score function for standard diffusion models, $\hat{s}_{\boldsymbol{\theta}}(\boldsymbol{x}, t) = \mathrm{diag}(\boldsymbol{\theta}^\ell)\boldsymbol{x}$, can remain inaccurate in the constrained coordinate.

Next, we consider the effective score function of diffusion models with guidance:

$$\hat{s}_{\boldsymbol{\theta}}^{\mathrm{g}}(\boldsymbol{x}, t) = \hat{s}_{\boldsymbol{\theta}}(\boldsymbol{x}, t) - \gamma \left\langle \frac{\boldsymbol{x} - \sigma_t \hat{s}_{\boldsymbol{\theta}}(\boldsymbol{x}, t)}{\alpha_t}, \boldsymbol{e}_d \right\rangle \boldsymbol{e}_d$$
$$= \sum_{j=1}^{d-1} \theta_j^\ell x_j \boldsymbol{e}_j + \left( \theta_d^\ell - \frac{\gamma}{\alpha_t} \left( 1 - \sigma_t \theta_d^\ell \right) \right) x_d \boldsymbol{e}_d.$$

Then, the last component is

$$\theta_d^{\ell, \mathrm{g}} := \left( 1 + \frac{\gamma \sigma_t}{\alpha_t} \right) \theta_d^\ell - \frac{\gamma}{\alpha_t}.$$

$\square$

This theorem shows that, for vanilla diffusion models, the learning error on the constrained coordinate remains large in gradient descent, leading to slow correction during sampling. Although guidance amplifies the update on the constrained dimension to reduce the sampling error caused by imperfect training of the score function, achieving $\theta_d^{\mathrm{g}} \approx \theta_d^* = -\sigma_t^{-2}$ at small $\sigma_t$ typically requires large $\gamma$, which is not reasonable in practice because it may introduce other instability issues. Moreover, this also requires very precise choices of $\gamma$ for different inference times $t$, which may require substantially more intensive hyperparameter tuning.

**Post-hoc distillation** Let $\widehat{x}_0(\epsilon)$ denote the teacher sample obtained by running an ODE solver (probability flow ODE) starting from noise $\epsilon \sim \mathcal{N}(\mathbf{0}, \mathbf{I})$ using a trained score model (with or without guidance). We distill this teacher into a student mapping

$$v_w(\epsilon) = \mathrm{diag}(w)\epsilon,$$

trained with a squared loss plus a hard-constraint penalty on the final coordinate:

$$\mathcal{L}_{\mathrm{distill}}(w) = \mathbb{E}_\epsilon\Big[\big\|v_w(\epsilon) - \widehat{x}_0(\epsilon)\big\|_2^2\Big] + \lambda\,\mathbb{E}_\epsilon\Big[\langle v_w(\epsilon), e_d\rangle^2\Big], \qquad \lambda \geq 0.$$

Then, the following theorem characterizes the solution of the distilled student learner and proves the constraint error achieved by the post-hoc distillation.

**Theorem B.2.** *Assume $\epsilon \sim \mathcal{N}(\mathbf{0}, \mathbf{I})$ and $v_w(\epsilon) = \mathrm{diag}(w)\epsilon$. Let $\widehat{x}_0(\epsilon)$ be any teacher output with $\mathbb{E}\|\widehat{x}_0(\epsilon)\|_2^2 < \infty$. Then $\mathcal{L}_{\mathrm{distill}}(w)$ is strictly convex and its unique minimizer satisfies, for each coordinate $j$,*

$$w_j^* = \frac{\mathbb{E}\big[\epsilon_j\,\widehat{x}_{0,j}(\epsilon)\big]}{1 + \lambda\,\mathbf{1}\{j = d\}}.$$

*In particular, the constrained coordinate is shrunk by a factor $1/(1 + \lambda)$:*

$$w_d^* = \frac{1}{1 + \lambda}\,\mathbb{E}\big[\epsilon_d\,\widehat{x}_{0,d}(\epsilon)\big], \qquad \mathbb{E}\big[\langle v_{w^*}(\epsilon), e_d\rangle^2\big] = (w_d^*)^2 \leq \frac{1}{(1 + \lambda)^2}\,\mathbb{E}\big[\widehat{x}_{0,d}(\epsilon)^2\big].$$

*Proof.* Expand the loss coordinate-wise. Since $v_w(\epsilon)_j = w_j\epsilon_j$,

$$\mathcal{L}_{\mathrm{distill}}(w) = \sum_{j=1}^d \mathbb{E}\big[(w_j\epsilon_j - \widehat{x}_{0,j})^2\big] + \lambda\,\mathbb{E}\big[(w_d\epsilon_d)^2\big].$$

Using $\mathbb{E}[\epsilon_j^2] = 1$, we get

$$\mathbb{E}\big[(w_j\epsilon_j - \widehat{x}_{0,j})^2\big] = w_j^2 - 2w_j\,\mathbb{E}[\epsilon_j\widehat{x}_{0,j}] + \mathbb{E}[\widehat{x}_{0,j}^2].$$

Thus, for $j \neq d$, the derivative is $2w_j - 2\mathbb{E}[\epsilon_j\widehat{x}_{0,j}]$, giving $w_j^* = \mathbb{E}[\epsilon_j\widehat{x}_{0,j}]$. For $j = d$, the coefficient of $w_d^2$ becomes $1 + \lambda$, giving $w_d^* = \mathbb{E}[\epsilon_d\widehat{x}_{0,d}]/(1 + \lambda)$. The bound on the constraint second moment follows by direct substitution. $\qquad\square$

**Remark.** Based on the above theorem, we can see that even if the teacher $\widehat{x}_0(\epsilon)$ has a nonzero constraint violation in the $d$-th coordinate due to imperfect score learning and/or numerical solver error, the distilled student, through the proposed distillation loss function, can systematically reduce this violation through the penalty parameter $\lambda$. Moreover, the penalization is applied only along one constrained direction, so learning on the remaining directions is still maintained, and the overall generation quality need not be degraded.

## C. Datasets

We consider the following widely used PDE benchmarks. Each dataset contains paired solution and coefficient fields defined on a $128 \times 128$ grid. These datasets are readily accessible from FNO (Li et al., 2020), DiffusionPDE (Huang et al., 2024a), and ECI-Sampling (Cheng et al., 2025).

### C.1. Darcy Flow

We adopt the Darcy Flow setup introduced in DiffusionPDE (Huang et al., 2024a), with the dataset released by FNO (Li et al., 2020). For completeness, we describe the generation process here. Specifically, we consider the steady-state Darcy flow equation on a 2D rectangular domain $\Omega \subset \mathbb{R}^2$ with no-slip boundary conditions:

$$-\nabla \cdot (a(c)\nabla u(c)) = q(c), \quad c \in \Omega, \quad u(c) = 0, \quad c \in \partial\Omega.$$

Here, $a(c)$ is the spatially varying permeability field with binary values, and $q(c)$ is set to 1 for constant forcing. The pair $(u, a)$ is jointly modeled by the diffusion model.

## C.2. Inhomogeneous Helmholtz Equation and Poisson Equation

We adopt the setup introduced in DiffusionPDE (Huang et al., 2024a), with the dataset released by FNO (Li et al., 2020). For completeness, we describe the generation process here. As a special case of the inhomogeneous Helmholtz equation, the Poisson equation is obtained by setting $k = 0$:

$$\nabla^2 u(c) = a(c), \quad c \in \Omega, \quad u(c) = 0, \quad c \in \partial\Omega.$$

Here, $a(c)$ is a piecewise constant forcing function. The pair $(u, a)$ is jointly modeled by the diffusion model.

## C.3. Burgers' Equation

We adopt the Burgers' Equation setup introduced in DiffusionPDE (Huang et al., 2024a), with the dataset released by FNO (Li et al., 2020). For completeness, we describe the generation process here. We study the 1D viscous Burgers' equation with periodic boundary conditions on a spatial domain $\Omega = (0, 1)$ and temporal domain $\tau \in (0, T]$:

$$\partial_\tau u(c, \tau) + \partial_c \left( \frac{u^2(c, \tau)}{2} \right) = \nu \partial_{cc}^2 u(c, \tau), \quad u(c, 0) = a(c), \quad c \in \Omega.$$

In our experiments, we set $\nu = 0.01$. Specifically, we use 128 temporal steps, where each trajectory has shape $128 \times 128$. The pair $(u, a)$ is jointly modeled by the diffusion model.

## C.4. Stokes Problem

We adopt the Stokes problem setup introduced in ECI-Sampling (Cheng et al., 2025) and use their released generation code. For completeness, we describe the generation process below.

The 1D Stokes problem is governed by the heat equation:

$$u_t = \nu u_{xx}, \quad x \in [0, 1], \ t \in [0, 1],$$

with the following boundary and initial conditions:

$$u(x, 0) = Ae^{-kx} \cos(kx), \quad x \in [0, 1], \quad u(0, t) = A \cos(\omega t), \quad t \in [0, 1],$$

where $\nu \geq 0$ is the viscosity, $A > 0$ is the amplitude, $\omega$ is the oscillation frequency, and $k = \sqrt{\omega/(2\nu)}$ controls the spatial decay. The analytical solution is given by:

$$u_{\text{exact}}(x, t) = Ae^{-kx} \cos(kx - \omega t).$$

In our experiments, we set $A = 2$, $k = 5$ and take $a := \omega \sim \mathcal{U}[2, 8]$ as the coefficient field to jointly model with $u$.

## C.5. Heat Equation

We adopt the heat equation setup introduced in ECI-Sampling (Cheng et al., 2025) and use their released generation code. For completeness, we describe the generation process below.

The 1D heat (diffusion) equation with periodic boundary conditions is defined as:

$$u_t = \alpha u_{xx}, \quad x \in [0, 2\pi], \ t \in [0, 1],$$

with the initial and boundary conditions:

$$u(x, 0) = \sin(x + \varphi), \quad u(0, t) = u(2\pi, t).$$

Here, $\alpha$ denotes the diffusion coefficient and $\varphi$ controls the phase of the sinusoidal initial condition. The exact solution is:

$$u_{\text{exact}}(x, t) = e^{-\alpha t} \sin(x + \varphi).$$

In our experiments, we set $\alpha = 3$ and take $a := \varphi \sim \mathcal{U}[0, \pi]$ as the coefficient to jointly model with $u$.

## C.6. Navier–Stokes Equation

We adopt the 2D Navier–Stokes (NS) setup from ECI-Sampling (Cheng et al., 2025) and use their released generation code. The NS equation in vorticity form for an incompressible fluid with periodic boundary conditions is given as:

$$\partial_t w(x,t) + u(x,t) \cdot \nabla w(x,t) = \nu \Delta w(x,t) + f(x), \quad x \in [0,1]^2, \, t \in [0,T],$$
$$\nabla \cdot u(x,t) = 0, \quad x \in [0,1]^2, \, t \in [0,T],$$
$$w(x,0) = w_0(x), \quad x \in [0,1]^2.$$

Here, $u$ denotes the velocity field and $w = \nabla \times u$ is the vorticity. The initial vorticity $w_0$ is sampled from $\mathcal{N}(0, 7^{3/2}(-\Delta + 49I)^{-5/2})$, and the forcing term is defined as $f(x) = 0.1\sqrt{2}\sin(2\pi(x_1 + x_2) + \phi)$, where $\phi \sim \mathcal{U}[0, \pi/2]$. We take $a := w_0$ as the coefficient to jointly model with $u$.

## C.7. Porous Medium Equation

We use the Porous Medium Equation (PME) setup provided by ECI-Sampling (Cheng et al., 2025), with zero initial and time-varying Dirichlet left boundary conditions:

$$u_t = \nabla \cdot (u^m \nabla u), \quad x \in [0,1], \, t \in [0,1],$$
$$u(x,0) = 0, \quad x \in [0,1],$$
$$u(0,t) = (mt)^{1/m}, \quad t \in [0,1],$$
$$u(1,t) = 0, \quad t \in [0,1].$$

The exact solution is $u_{\text{exact}}(x,t) = (m \cdot \text{ReLU}(t - x))^{1/m}$. The exponent $m$ is sampled from $\mathcal{U}[1,5]$. We take $a := m$ as the coefficient to jointly model with $u$.

## C.8. Stefan Problem

We also adopt the Stefan problem configuration from ECI-Sampling (Cheng et al., 2025), which is a nonlinear case of the Generalized Porous Medium Equation (GPME) with fixed Dirichlet boundary conditions:

$$u_t = \nabla \cdot (k(u)\nabla u), \quad x \in [0,1], \, t \in [0,T],$$
$$u(x,0) = 0, \quad x \in [0,1],$$
$$u(0,t) = 1, \quad t \in [0,T],$$
$$u(1,t) = 0, \quad t \in [0,T],$$

where $k(u)$ is a step function defined by a shock value $u^*$:

$$k(u) = \begin{cases} 1, & u \geq u^*, \\ 0, & u < u^*. \end{cases}$$

The exact solution is:

$$u_{\text{exact}}(x,t) = \mathbb{1}_{[u \geq u^*]}\left(1 - (1 - u^*)\frac{\text{erf}(x/(2\sqrt{t}))}{\text{erf}(\alpha)}\right),$$

where $\alpha$ satisfies the nonlinear equation $(1 - u^*)/\sqrt{\pi} = u^* \text{erf}(\alpha)\alpha \exp(\alpha^2)$. We follow ECI-Sampling to take $a := u^* \sim \mathcal{U}[0.55, 0.7]$ as the coefficient to jointly model with $u$.

# D. Experimental Setup

This section provides details on the model architecture, training configurations for diffusion and distillation, evaluation protocols, and baseline methods.

## D.1. Model Structure

We follow ECI-sampling (Cheng et al., 2025) and adopt the Fourier Neural Operator (FNO) (Li et al., 2020) as both the teacher diffusion model and the student distillation model. A sinusoidal positional encoding (Vaswani et al., 2017) is appended as an additional input dimension. Specifically, we use a four-layer FNO with a frequency cutoff of $32 \times 32$, a time embedding dimension of 32, a hidden channel width of 64, and a projection dimension of 256.

## D.2. Diffusion and Distillation Training Setup

For diffusion training, we employ a standard linear noise schedule (Liu et al., 2023b;a; Lipman et al., 2023; Liu et al., 2023c) with a batch size of 128 and a total of 10,000 iterations. The model is optimized using Adam (Kingma & Ba, 2014) with a learning rate of $3 \times 10^{-2}$.

During distillation, we use Euler's method with 100 uniformly spaced timesteps from $t = 1$ to $t = 0$ for sampling. Every 100 epochs, we resample 1024 new noise–data pairs for supervision. Distillation is trained for 2000 epochs using Adam (learning rate $3 \times 10^{-2}$), with early stopping based on the squared norm of the observation loss, i.e., $\|d_{\theta'}(\varepsilon) - x\|^2$.

The physics constraint weight $\lambda_{\text{train}}$ is set to 10 for Darcy Flow, Burgers' Equation, Stokes Problem, Heat Equation, Navier–Stokes, Porous Medium Equation, and Stefan Problem. For Helmholtz and Poisson equations, we increase $\lambda_{\text{train}}$ to $10^6$ due to the stiffness of these PDEs. All experiments are conducted on an NVIDIA RTX 4090 GPU.

## D.3. Evaluation Setup

For physics-based data simulation, we evaluate models with and without physics refinement: the number of gradient-based refinement steps $N$ is set to 0 or 50. The step size $\eta$ is aligned with the dataset-specific $\lambda_{\text{train}}$ used during distillation.

In forward and inverse problems, the observation mask $M$ defines the known entries. For forward problems, the mask has ones at boundary entries. For partial reconstruction, the mask is sampled randomly with 20% of entries set to 1 (observed), and the rest to 0 (missing). All evaluations are conducted on an NVIDIA RTX 4090 GPU.

## D.4. Baseline Methods

We describe the configurations of all baseline methods used for comparison. Where necessary, we adapt our diffusion training and sampling codebase to implement their respective constraint mechanisms.

**ECI-sampling.** We follow the approach of directly substituting hard constraints into the posterior mean $\mathbb{E}[x_0 \mid x_t]$ based on a predefined observation mask. Specifically, we project these constraints at each DDIM step (Song et al., 2021) using a correction operator $C$:

$$\boldsymbol{x}_{t-dt} = C(\hat{\boldsymbol{x}}_\theta(\boldsymbol{x}_t, t)) \cdot (1 - t + \mathrm{d}t) + (\boldsymbol{x}_t - \hat{\boldsymbol{x}}_\theta(\boldsymbol{x}_t, t)) \cdot (t - \mathrm{d}t), \tag{12}$$

where $t$ flows backward from 1 to 0, and $\hat{\boldsymbol{x}}_\theta$ denotes the posterior mean estimated using Tweedie's formula.

**DiffusionPDE.** This method employs diffusion posterior sampling (DPS) (Chung et al., 2023), where each intermediate sample $\boldsymbol{x}_t$ is guided by the gradient of the PDE residual evaluated on the posterior mean:

$$\boldsymbol{x}_{t-dt} = \boldsymbol{x}_t + v_\theta(\boldsymbol{x}_t, t) \cdot \mathrm{d}t - \eta_t \nabla_{\boldsymbol{x}_t} \|\mathcal{R}(\hat{\boldsymbol{x}}_\theta(\boldsymbol{x}_t, t))\|^2, \tag{13}$$

where $v_\theta(\boldsymbol{x}_t, t)$ is the learned velocity field from the reverse-time ODE sampler, and $\eta_t$ is a hyperparameter. In our experiments, we set $\eta_t$ equal to $\lambda_{\text{train}}$ for each dataset.

**PIDM.** This method incorporates an additional residual loss into the diffusion training objective, evaluated on the posterior mean $\mathbb{E}[x_0 \mid x_t]$. Specifically, PIDM (Bastek et al., 2025) augments the standard diffusion loss with a physics-based term:

$$\mathcal{L}_{\text{PIDM}} = \mathcal{L}_{\text{diffusion}} + \lambda_t \|\mathcal{R}(\hat{\boldsymbol{x}}_\theta(\boldsymbol{x}_t, t))\|^2, \tag{14}$$

where $\mathcal{L}_{\text{diffusion}}$ is the original diffusion training loss, and $\lambda_t$ is the residual loss weight. In our experiments, we set $\lambda_t$ to $\lambda_{\text{train}}$ for each dataset as it gives stable performance.

**D-Flow.** For this standard method (Ben-Hamu et al., 2024), we build on the official implementation of ECI-sampling (Cheng et al., 2025) and introduce an additional PDE residual loss evaluated on the final sample. The weighting $\lambda_{\text{train}}$ is aligned with

our setup across datasets. Specifically, the implementation follows the D-Flow setup in ECI-sampling (Cheng et al., 2025): we discretize the sampling trajectory into 100 denoising steps and perform gradient-based optimization on the input noise over 50 iterations to minimize the physics residual loss. At each iteration, gradients are backpropagated through the entire 100-step trajectory, resulting in a total of 5,000 function evaluations (NFE) per sample. This leads to significantly higher computational cost compared to our one-step method.

**Teacher.** This baseline refers to sampling directly from the trained teacher diffusion model without incorporating any PDE-based constraint or guidance mechanism.

## E. Generative Evaluations on More Datasets

In this section, we report results on more datasets and compare them with other baseline methods, as shown in Table 5. PIDDM consistently improves over the baselines, especially in physics residual error.

*Table 5.* Generative metrics on various constrained PDEs. PDE error is the MSE of the evaluated physics residual error. The best results are in **bold**.

| Dataset | Metric | PIDDM-1 | PIDDM-ref | ECI | DiffusionPDE | D-Flow | PIDM | FM |
|---|---|---|---|---|---|---|---|---|
| Helmholtz | MMSE ($\times 10^{-1}$) | 0.265 | **0.185** | 0.318 | 0.335 | 0.140 | 0.352 | 0.296 |
| | SMSE ($\times 10^{-1}$) | 0.195 | **0.169** | 0.289 | 0.301 | 0.106 | 0.325 | 0.210 |
| | PDE Error ($\times 10^{-9}$) | 0.054 | **0.034** | 2.135 | 1.812 | 0.680 | 1.142 | 2.104 |
| | NFE ($\times 10^{3}$) | **0.001** | 0.100 | 0.500 | 0.100 | 5.000 | 0.100 | 0.100 |
| Stokes Problem | MMSE ($\times 10^{-2}$) | 0.298 | **0.182** | 0.335 | 0.342 | 0.301 | 0.361 | 0.310 |
| | SMSE ($\times 10^{-2}$) | 0.425 | **0.312** | 0.455 | 0.469 | 0.441 | 0.484 | 0.430 |
| | PDE Error ($\times 10^{-3}$) | 0.241 | **0.194** | 0.585 | 0.498 | 0.318 | 0.432 | 0.578 |
| | NFE ($\times 10^{3}$) | **0.001** | 0.100 | 0.500 | 0.100 | 5.000 | 0.100 | 0.100 |
| Heat Equation | MMSE ($\times 10^{-3}$) | 0.901 | **0.845** | 4.620 | 4.600 | 1.452 | 4.580 | 4.544 |
| | SMSE ($\times 10^{-2}$) | 0.816 | **0.790** | 1.612 | 1.598 | 0.892 | 1.587 | 1.565 |
| | PDE Error ($\times 10^{-5}$) | 3.265 | **2.910** | 4.120 | 4.100 | 3.698 | 4.150 | 4.354 |
| | NFE ($\times 10^{3}$) | **0.001** | 0.100 | 0.500 | 0.100 | 5.000 | 0.100 | 0.100 |
| Navier–Stokes Equation | MMSE ($\times 10^{-4}$) | 0.285 | **0.264** | 0.302 | 0.299 | 0.288 | 0.306 | 0.294 |
| | SMSE ($\times 10^{-4}$) | 0.218 | **0.210** | 0.323 | 0.321 | 0.225 | 0.327 | 0.314 |
| | PDE Error ($\times 10^{-5}$) | 3.184 | **2.945** | 6.910 | 6.740 | 3.200 | 6.950 | 7.222 |
| | NFE ($\times 10^{3}$) | **0.001** | 0.100 | 0.500 | 0.100 | 5.000 | 0.100 | 0.100 |
| Porous Medium Equation | MMSE ($\times 10^{-3}$) | 4.555 | **4.210** | 7.742 | 7.698 | 5.203 | 7.762 | 7.863 |
| | SMSE ($\times 10^{-1}$) | 2.143 | **2.051** | 2.573 | 2.602 | 2.327 | 2.589 | 2.639 |
| | PDE Error ($\times 10^{-5}$) | 3.412 | **3.110** | 4.982 | 4.945 | 3.548 | 4.917 | 5.523 |
| | NFE ($\times 10^{3}$) | **0.001** | 0.100 | 0.500 | 0.100 | 5.000 | 0.100 | 0.100 |
| Stefan Problem | MMSE ($\times 10^{-3}$) | 0.231 | **0.220** | 0.248 | 0.249 | 0.238 | 0.252 | 0.245 |
| | SMSE ($\times 10^{-3}$) | 0.278 | **0.268** | 0.315 | 0.318 | 0.289 | 0.320 | 0.307 |
| | PDE Error ($\times 10^{-2}$) | 0.081 | **0.070** | 0.410 | 0.398 | 0.095 | 0.405 | 0.458 |
| | NFE ($\times 10^{3}$) | **0.001** | 0.100 | 0.500 | 0.100 | 5.000 | 0.100 | 0.100 |

## F. Qualitative Results on the Darcy Forward Problem

Figure 4 compares the predicted Darcy pressure fields and their corresponding data- and PDE-error maps for each baseline and for our PIDDM. DiffusionPDE and ECI reproduce the coarse flow pattern but exhibit large pointwise errors and pronounced residual bands. In contrast, PIDDM produces the visually sharpest solution and the lowest error intensities in both maps, confirming the quantitative gains reported in the main text.

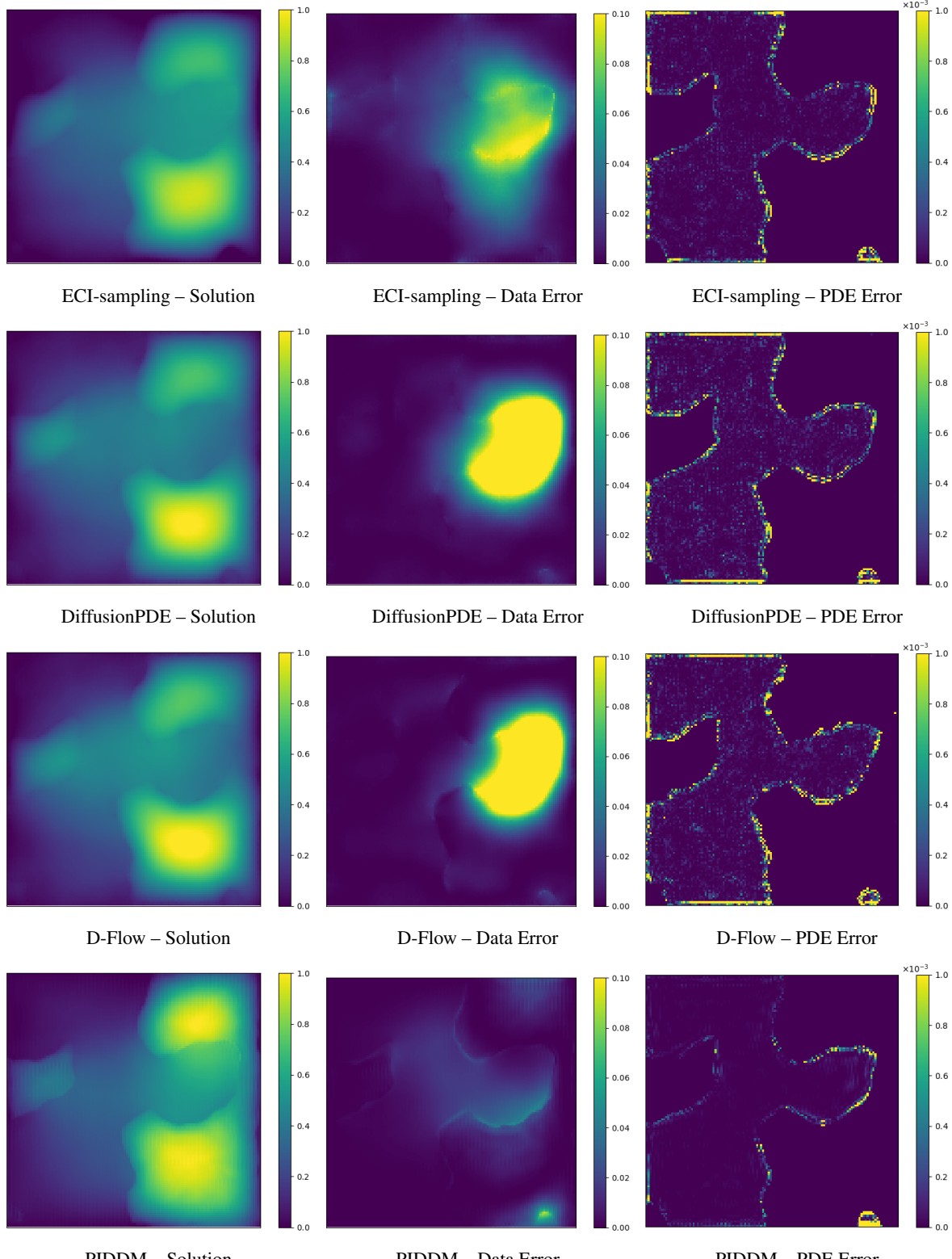

*Figure 4.* Qualitative comparison on the Darcy *forward* problem. Each column shows (left) the predicted solution field, (middle) pointwise data error, and (right) PDE residual error. Our PIDDM (bottom row) delivers visibly lower data and PDE errors than other baselines while maintaining sharp solution details.

# G. Additional Experiments

## G.1. Additional Review-Driven Ablations

We include additional experiments requested during review to clarify component attribution, diversity preservation, teacher-quality sensitivity, architecture robustness, dynamic loss weighting, and latent-optimization stability. These experiments use the same evaluation protocol as the main text and are reported on Darcy unless otherwise specified. The compute-matched comparison is reported in Table 3.

**Component attribution.** Table 6 separates the effects of one-step distillation and final-sample PDE supervision. Distillation alone mainly provides the efficiency gain but does not improve PDE satisfaction, whereas final-sample PDE supervision is the key component for physical consistency. PIDDM combines both effects and achieves a better quality–compute trade-off.

**Diversity preservation.** Beyond MMSE, SMSE, and FPD, we report sliced Wasserstein distance (SWD) and average pairwise distance difference (APDD). For samples $x^{(1)}, \ldots, x^{(N)}$, define

$$\text{APD} = \frac{2}{N(N-1)} \sum_{i<j} \|x_i - x_j\|_2, \qquad \text{APDD} = \frac{|\text{APD}_S - \text{APD}_T|}{\text{APD}_T},$$

where $\text{APD}_S$ and $\text{APD}_T$ are computed on student and teacher samples, respectively. Table 7 shows that PIDDM preserves diversity better than vanilla distillation, with PIDDM-ref performing best.

**Teacher quality and architecture robustness.** Table 8 shows that student performance improves with teacher checkpoint quality early in training, but gains become marginal after 6k teacher iterations. Table 9 further verifies that PIDDM improves physical consistency over DiffusionPDE under U-Net and CNN auto-encoder backbones, showing that the final-sample supervision strategy is not tied to FNO.

**Dynamic weighting and latent optimization stability.** For stiff systems such as Helmholtz and Poisson, raw PDE residual magnitudes can differ by orders of magnitude across PDE families. We therefore evaluate a dynamic weighting variant that rescales the PDE term by a detached running residual magnitude to target a comparable scale of $10^{-4}$. Table 10 shows comparable performance to manually tuned weights. Tables 11 and 12 show that the downstream latent optimization converges quickly and has lower variance across random initializations than DiffusionPDE.

**Final-sample PDE supervision across distillation backbones.** Table 13 compares standard distillation backbones with and without final-sample PDE supervision (PS). Removing PS only slightly changes MSE but increases PDE error by about one order of magnitude, indicating that the backbone mainly affects one-step distillation quality while final-sample PDE supervision drives the physics-consistency gain.

*Table 6.* Component attribution on Darcy generation. Distill denotes pure one-step distillation without final-sample PDE supervision.

| Metric | PIDDM-ref | Distill | Teacher | D-Flow |
|---|---|---|---|---|
| MMSE ($\times 10^{-2}$) | 0.037 | 0.121 | 0.108 | 0.129 |
| SMSE ($\times 10^{-2}$) | 0.002 | 0.102 | 0.069 | 0.085 |
| PDE error ($\times 10^{-4}$) | 0.148 | 1.642 | 1.585 | 0.532 |
| FPD | 0.385 | 0.825 | 0.782 | 0.995 |
| NFE | 80 | 1 | 100 | 5000 |

*Table 7.* Diversity preservation on Darcy generation. Lower SWD and APDD indicate closer distributional diversity to the reference samples.

| Metric | PIDDM-1 | PIDDM-ref | Distillation | Teacher |
|---|---|---|---|---|
| SWD | 0.058 | 0.032 | 0.103 | 0.089 |
| APDD | 0.050 | 0.032 | 0.067 | 0.034 |

*Table 8.* Sensitivity to teacher checkpoint quality on Darcy generation.

| Metric | 2k | 4k | 6k | 8k | 10k |
|---|---|---|---|---|---|
| MMSE ($\times 10^{-2}$) | 0.174 | 0.136 | 0.121 | 0.119 | 0.112 |
| SMSE ($\times 10^{-2}$) | 0.124 | 0.098 | 0.085 | 0.085 | 0.082 |
| PDE error ($\times 10^{-4}$) | 0.336 | 0.247 | 0.231 | 0.229 | 0.226 |

*Table 9.* Architecture robustness on Darcy generation. PIDDM improves over posterior-mean guidance when using U-Net and CNN auto-encoder backbones.

| Metric | PIDDM(UNet) | DiffusionPDE(UNet) | PIDDM(AE) | DiffusionPDE(AE) |
|---|---|---|---|---|
| MMSE ($\times 10^{-2}$) | 0.243 | 0.753 | 0.310 | 0.975 |
| SMSE ($\times 10^{-2}$) | 0.211 | 0.431 | 0.257 | 0.692 |
| PDE error ($\times 10^{-4}$) | 0.743 | 1.566 | 0.902 | 1.848 |
| FPD | 0.948 | 1.651 | 1.182 | 1.906 |

*Table 10.* Dynamic weighting for stiff PDEs. Dynamic weighting rescales the PDE term by a detached running residual magnitude, targeting a comparable residual scale of $10^{-4}$.

| Metric | Helmholtz(dynamic) | Helmholtz(manual) | Poisson(dynamic) | Poisson(manual) |
|---|---|---|---|---|
| MMSE ($\times 10^{-1}$) | 0.257 | 0.265 | 0.017 | 0.016 |
| SMSE ($\times 10^{-1}$) | 0.184 | 0.195 | 0.035 | 0.036 |
| PDE error ($\times 10^{-9}$) | 0.053 | 0.054 | 0.069 | 0.073 |

*Table 11.* Convergence of latent optimization on the Darcy forward task.

| Metric | 10 | 20 | 40 | 60 | 80 |
|---|---|---|---|---|---|
| PDE loss ($\times 10^{-4}$) | 0.727 | 0.173 | 0.152 | 0.149 | 0.145 |
| Data MSE | 0.119 | 0.012 | 0.003 | 0.002 | 0.002 |

*Table 12.* Error variance across random initializations. Values are MSE variance ($\times 10^{-5}$).

| Method | Forward | Inverse | Reconstruction |
|---|---|---|---|
| PIDDM-ref | 8.57 | 9.49 | 8.14 |
| DiffusionPDE | 46.39 | 54.61 | 42.72 |

*Table 13.* Effect of final-sample PDE supervision (PS) across one-step distillation backbones on the Darcy forward task.

| Metric | RF-2+PS | RF-2 w/o PS | DMD+PS | DMD w/o PS | CM+PS | CM w/o PS | DiffusionPDE | PIDM |
|---|---|---|---|---|---|---|---|---|
| MSE ($\times 10^{-1}$) | 0.127 | 0.143 | 0.255 | 0.261 | 0.283 | 0.291 | 0.691 | 0.380 |
| PDE error ($\times 10^{-4}$) | 0.098 | 1.304 | 0.134 | 1.248 | 0.083 | 1.373 | 1.576 | 1.248 |

### G.2. Correlated MoG Constraint Satisfaction

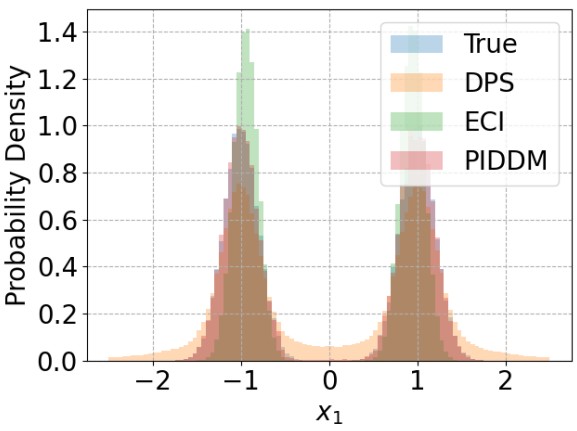 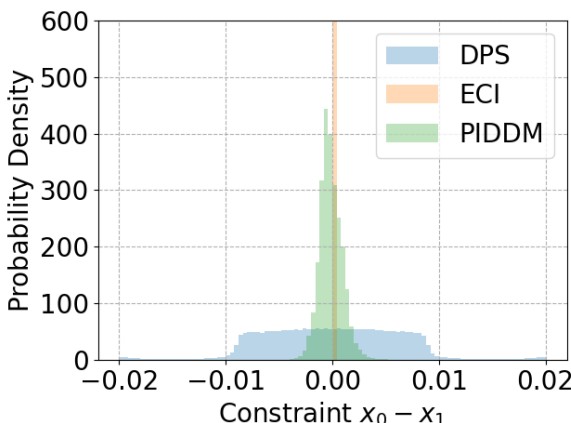

*(a)* Marginal distribution over $x_1$

*(b)* Constraint deviation $x_0 - x_1$

*Figure 5.* Constraint satisfaction on correlated MoG. Comparison of generated samples using DPS, ECI, and PIDDM. PIDDM closely matches the target distribution while satisfying constraints.

We investigate a controlled Mixture-of-Gaussians (MoG) setting to evaluate constraint satisfaction in generative models. The target distribution is a correlated, two-component Gaussian mixture:

$$p_{\text{MoG}}(\mathbf{x}) = \tfrac{1}{2}\mathcal{N}\left(\mathbf{x}; [-1, -1]^\top, \Sigma\right) + \tfrac{1}{2}\mathcal{N}\left(\mathbf{x}; [+1, +1]^\top, \Sigma\right), \tag{15}$$

where

$$\Sigma = \sigma^2 \begin{bmatrix} 1 & \rho \\ \rho & 1 \end{bmatrix}, \qquad \sigma^2 = 0.04, \quad \rho = 0.99999.$$

The high correlation $\rho = 0.99999$ ensures that the analytic score function $\nabla_\mathbf{x} \log p_{\text{MoG}}(\mathbf{x})$ remains well-defined, despite the near-singular covariance. The physical constraint is defined as:

$$\mathcal{F}(\boldsymbol{x}) = |\boldsymbol{x}_0 - \boldsymbol{x}_1|^2 = 0. \tag{16}$$

**Baselines.** DPS and ECI both integrate the analytical score using 1000-step Euler discretization over $(0, 1)$. DPS applies constraint guidance via gradient descent on $\mathcal{F}(\boldsymbol{x})$ at each step, using a loss weight of 300. ECI enforces the constraint by directly projecting the posterior mean to satisfy $\mathcal{F}(\boldsymbol{x}) = 0$.

**PIDDM.** A teacher diffusion model is constructed using a probability-flow ODE with 100-step Euler integration, leveraging the analytic score. It generates 50,000 training pairs $(\boldsymbol{\varepsilon}, \boldsymbol{x}_0)$ that are used to train a one-step student model, a ReLU-activated MLP with two hidden layers (100 neurons each) via the loss:

$$\mathcal{L}_{\text{train}} = \frac{1}{N} \sum_{i=1}^{N} |d_{\boldsymbol{\theta}}(\boldsymbol{\varepsilon}_i) - \boldsymbol{x}_{0,i}|^2 + \lambda_{\text{train}} \mathcal{F}(d_{\boldsymbol{\theta}}(\boldsymbol{\varepsilon}_i)), \quad \lambda_{\text{train}} = 1. \tag{17}$$

Training uses Adam optimizer (lr $= 10^{-3}$, batch size $= 2048$). During inference, latent noise $\boldsymbol{\varepsilon}$ is optimized via 80 steps of LBFGS with strong-Wolfe line search, learning rate $3 \times 10^{-3}$, and gradient tolerance $10^{-7}$, with $\lambda_{\text{infer}} = 1$.

**Results.** Figure 5(a) shows that all methods recover the bimodal structure of $\boldsymbol{x}_1$. However, as shown in Figure 5(b), DPS fails to fully satisfy the constraint, with $\boldsymbol{x}_0 - \boldsymbol{x}_1$ spread over $\pm 10^{-2}$, while ECI enforces it exactly but distorts the marginal distribution. In contrast, PIDDM maintains both constraint satisfaction (standard deviation $\approx 2 \times 10^{-3}$) and distributional fidelity.

