# OpenReview forum: "Physics-Informed Distillation of Diffusion Models for PDE-Constrained Generation"
_ICML.cc/2026/Conference — ICML 2026 regular_

### Official Review · Reviewer_xvqZ · 2026-02-17

**Soundness:** 2
**Presentation:** 2
**Significance:** 2
**Originality:** 2
**Overall Recommendation:** 4
**Confidence:** 3

**Summary:**

The authors propose PIDDM, a method targeted for solving inverse problems arising in PDE with diffusion(flow) models. In the training stage, the authors additionally incorporate physics-informed loss on top of the distillation training loss. During inference, standard gradient-based guidance is performed with the trained model.

**Compliance With Llm Reviewing Policy:**

Affirmed.

**Final Justification:**

All concerns have been resolved.

**Key Questions For Authors:**

1. Can the authors comment on the trade-off between using the posterior sample and the posterior mean in terms of the accuracy of approximation and the variance?

2. Please clarify the key components to improve distillation training.

**Limitations:**

Yes

**Strengths And Weaknesses:**

### Strenghts

1. Solving inverse problems in the PDE domain is an important problem.

2. The proposed method is sound. One-step methods such as meanflow would be able to produce posterior samples rather than the posterior mean, which could lead to better approximations.

3. Empirical results show improved performance.

### Weaknesses

1. The approximation of the measurement consistency guidance *may* improve when you have a posterior sample rather than the posterior mean. However, notice that for the approximation to be exact, you have to integrate over **all the possible** posterior samples, not a single sample. Hence, you cannot guarantee that you have a better estimate when you are using a single posterior sample, as you would have much higher variance.

2. It is widely known in the literature that the Jensen gap exists. The question is whether using single posterior samples along the trajectory actually mitigates this. Currently, figure 2 does not show this.

3. Currently, it is hard to understand which component of the distillation is crucial, and why the well-established distillation methods cannot be used directly. Table 3 seems to be comparing against using DMD, CM, etc. *on top of* existing components, which is a bit confusing to comprehend.

---

> ### Author Rebuttal · Authors · 2026-03-31
>
> Q1: The approximation of the measurement consistency guidance may improve when you have a posterior sample rather than the posterior mean. However, notice that for the approximation to be exact, you have to integrate over all the possible posterior samples, not a single sample. Hence, you cannot guarantee that you have a better estimate when you are using a single posterior sample, as you would have much higher variance. Can the authors comment on the trade-off between using the posterior sample and the posterior mean in terms of the accuracy of approximation and the variance?
>
> We thank the reviewer for the insightful question. We would like to clarify that **our goal is not to better estimate the conditional score using a posterior sample instead of a posterior mean**. In fact, we fully agree with the reviewer’ that exact conditional guidance would require integrating over all posterior samples rather than using a single sample, which is generally intractable. Precisely for this reason, we do not attempt to modify or improve measurement-consistency guidance inside the teacher’s denoising process. Instead, PIDDM moves physics enforcement from guided diffusion sampling to post-hoc distillation, where the PDE constraint is applied directly to the final output of a one-step student. Therefore, the posterior-sample-versus-posterior-mean trade-off in approximating conditional guidance is not the object of our method; our claim is narrower, namely improved sample-level physical consistency of the final generated output. We will revise the wording to make this scope explicit.
>
> Q2: It is widely known in the literature that the Jensen gap exists. The question is whether using single posterior samples along the trajectory actually mitigates this. Currently, figure 2 does not show this.
>
> We thank the reviewer for pointing this out. To clarify, **we do not claim that single posterior samples along the denoising trajectory mitigate Jensen’s Gap**. As clarified in Q1, that is not what PIDDM does. Figure 2's role is only to motivate and illustrate the Jensen' gap mismatch introduced by posterior mean-based PDE enforcement in prior diffusion methods in the PDE scenario, where the relevant inconsistency is $R(\mathbb{E}[x_0|x_t]) \neq \mathbb{E}[R(x_0)|x_t]$. In contrast, PIDDM does not modify trajectory-level guided sampling during denoising; instead, it moves physics enforcement to post-hoc distillation and applies the PDE constraint directly to the final output of a one-step student, $R(d(\epsilon))$. We will revise the text around Figure 2 to make this explicit.
>
> Q3:Currently, it is hard to understand which component of the distillation is crucial, and why the well-established distillation methods cannot be used directly. Table 3 seems to be comparing against using DMD, CM, etc. on top of existing components, which is a bit confusing to comprehend. Please clarify the key components to improve distillation training.
>
> We thank the reviewer for raising this point. To clarify, **the crucial component is the final-sample PDE supervision during one-step distillation**. The specific distillation backbone is largely orthogonal to this idea. Therefore, our intention in Table 3 is precisely intended to show **direct compatibility with standard one-step distillation methods**, not to present DMD/CM as separate competing baselines. We will revise the presentation to make this distinction clearer.
>
> To isolate the role of final-sample PDE supervision (PS), we compare the same distillation backbone with and without PS on the Darcy forward task. For reference, we also include two representative PDE-guided baselines (DiffusionPDE and PIDM).
>
> ||+RF-2 + PS|+RF-2 w/o PS|DMD + PS|DMD w/o PS|CM + PS|CM w/o PS|DiffusionPDE|PIDM|
> |-|-|-|-|-|-|-|-|-|
> |MSE 1e-1|0.127|0.143|0.255|0.261|0.283|0.291|0.691|0.380|
> |PDE Error 1e-4|0.098|1.304|0.134|1.248|0.083|1.373|1.576|1.248|
>
> These results show a consistent pattern across all three distillation backbones: removing PS only slightly changes MSE, but increases PDE error by about one order of magnitude. Moreover, **without PS, the PDE error becomes comparable to or worse than representative PDE-guided baselines such as DiffusionPDE and PIDM**. This indicates that the backbone mainly affects the quality of one-step distillation, whereas **final-sample PDE supervision is the component for the physics-consistency gain**. In other words, standard one-step distillation provides the speed-up, while PIDDM turns distillation into a training objective for final-sample physical consistency. We will revise Table 3 and the surrounding text to explicitly separate **(i)** backbone choice and **(ii)** the presence/absence of final-sample PDE supervision, so that the key contribution is easier to interpret.

---

> > ### Author Rebuttal · Reviewer_xvqZ · 2026-04-01
> >
> > I thank the authors for the rebuttal. Most of my concerns have been addressed. I have a follow-up comment. The notion of Jensen gap within the context of solving inverse problems with diffusion models was introduced in [1], not [2]. The citations should be fixed accordingly.
> >
> > [1] Chung, Hyungjin, et al. "Diffusion posterior sampling for general noisy inverse problems." ICLR 2023.
> >
> > [2] Bastek, Jan-Hendrik, WaiChing Sun, and Dennis M. Kochmann. "Physics-informed diffusion models." ICLR 2025.

---

> > > ### Author Response · Authors · 2026-04-06
> > >
> > > Thank you for the helpful suggestion. We agree with this point and will revise the paper accordingly to make it clearer in the final version. In particular, we will revise our current statement on Jensen's gap (Line 89) to: "This mismatch, known as Jensen's gap (Chung et al., 2023 [1]) and originally identified in inverse problems, can lead to degraded physical fidelity." If this addresses your concern and there are no further questions, we would greatly appreciate it if you could reconsider the evaluation.
> > >
> > > [1] Chung, Hyungjin, et al. "Diffusion posterior sampling for general noisy inverse problems." ICLR 2023.

---

### Official Review · Reviewer_ze8L · 2026-03-08

**Soundness:** 3
**Presentation:** 3
**Significance:** 3
**Originality:** 3
**Overall Recommendation:** 4
**Confidence:** 2

**Summary:**

The authors address the Jensen's Gap in physics-informed diffusion models, which arises when evaluating partial differential equation constraints on the posterior mean during the denoising process. To bypass this issue, they propose a Physics-Informed Distillation framework. The method first trains a standard diffusion teacher model to generate noise-data pairs and subsequently distills a one-step student model. The distillation objective combines a standard reconstruction loss against the teacher's output with a direct physical residual penalty on the final generated sample. Finally, the authors evaluate the framework across several standard physical benchmarks for both unconditional generation and downstream tasks.

**Compliance With Llm Reviewing Policy:**

Affirmed.

**Ethical Review Concerns:**

yes

**Final Justification:**

The reply answered my question, so I'm inclined to keep the score.

**Key Questions For Authors:**

1. In Appendix D.2, the distillation loss weight is set to 10 for most datasets but is increased to one million for Helmholtz and Poisson equations . Could the authors elaborate on the underlying criteria for this significant adjustment and discuss how a practitioner should systematically determine this weight for an unseen PDE system? Providing a heuristic, dynamic weighting strategy, or scaling rule would help clarify the method's practical generalizability and its robustness against sub-optimal weight initializations.

2. For downstream tasks, Algorithm 3 relies on gradient-based optimization of the initial noise . While Table 2 indicates that this process requires 80 network function evaluations, the paper lacks a detailed analysis of the optimization's actual convergence behavior and stability. Given that the mapping from the latent space to the physical domain is typically non-convex, how do the authors ensure this optimization does not get trapped in local minima or suffer from gradient instability within these 80 iterations, especially in cases of sparse observations? Could the authors provide convergence curves or an analysis of variance across different random initializations to further substantiate the method's reliability?

3. Can you articulate the fundamental algorithmic novelty of this work beyond augmenting a standard one-step diffusion distillation objective with a physics residual penalty?

**Limitations:**

yes

**Strengths And Weaknesses:**

**Strengths:**

- The authors creatively decouple constraint enforcement from the iterative denoising process. Moving the physical constraint to a post-hoc distillation stage provides a conceptually clean solution that fundamentally bypasses the Jensen's Gap.
- Distilling the pre-trained diffusion model into a single-step student architecture drastically improves inference efficiency. The method successfully reduces the computational cost from thousands of network evaluations down to a single step for physical simulation tasks.
- The authors validate the framework across a comprehensive set of physical systems including Darcy, Poisson, and Burgers equations. The empirical results demonstrate competitive physical fidelity across various downstream tasks such as forward solving and partial reconstruction.

**Weaknesses:**

- The model exhibits sensitivity to the distillation loss weight, particularly when dealing with stiff equations. As noted in Appendix D.2, while a weight of 10 is sufficient for most datasets, it must be increased to one million for the Helmholtz and Poisson equations . The paper currently lacks a discussion on how to systematically determine this weight for unseen PDE systems, which might affect the method's generalization in practical applications.
- For downstream tasks like Algorithm 3, the model relies on gradient descent to optimize the initial latent noise to satisfy observation masks and PDE constraints. Since the mapping from the latent space to the physical domain is typically non-convex, directly optimizing the noise may encounter local optima or gradient instability, especially when dealing with sparse observations or high-dimensional noise spaces. The paper lacks a detailed analysis of the convergence speed and stability of this optimization process, particularly when it involves the mask mixing operation.
- The core algorithmic contribution lacks sufficient novelty for the main track of a premier machine learning conference. The method essentially applies a standard physics-informed penalty to an existing one-step diffusion distillation pipeline. While the engineering execution is sound, this direct combination offers limited fundamental theoretical innovation.

---

> ### Author Rebuttal · Authors · 2026-03-31
>
> Q: The model exhibits sensitivity to the distillation loss weight, particularly when dealing with stiff equations. As noted in Appendix D.2, while a weight of 10 is sufficient for most datasets, it must be increased to one million for the Helmholtz and Poisson equations . The paper currently lacks a discussion on how to systematically determine this weight for unseen PDE systems, which might affect the method's generalization in practical applications.
>
> We thank the reviewer for the helpful comment. We agree that the current paper should better explain how the PDE-loss weight is selected, especially for unseen systems. In our experiments, **the large numerical difference in the chosen weight mainly reflects scale differences in the raw PDE residual across PDE families, rather than a qualitatively different role of the penalty term**. In particular, for Helmholtz and Poisson, the residual magnitude is orders of magnitude smaller than for the other benchmarks (1e-9 vs 1e-4), so a numerically larger coefficient is needed to put the PDE term on a comparable scale to the reconstruction term during training.
>
> To make this more systematic, we add a **simple normalization / dynamic-weighting** rule that rescales the PDE loss by its running magnitude to 1e-4, which largely removes the need for manual retuning. We have run this variant on the affected benchmarks and obtained results comparable to manual tuning. We will add this discussion and the corresponding ablation to the revision.
>
> ||Helmholtz(dynamic)|Helmholtz (manual) |Poisson (dynamics)|Poisson(manual)|
> |-|-|-|-|-|
> |MMSE 1e-1|0.257|0.265|0.017|0.016|
> |SMSE 1e-1|0.184|0.195|0.035|0.036|
> |PDE 1e-9|0.053|0.054|0.069|0.073|
>
> Q: For downstream tasks like Algorithm 3, the model relies on gradient descent to optimize the initial latent noise to satisfy observation masks and PDE constraints. Since the mapping from the latent space to the physical domain is typically non-convex, directly optimizing the noise may encounter local optima or gradient instability, especially when dealing with sparse observations or high-dimensional noise spaces. The paper lacks a detailed analysis of the convergence speed and stability of this optimization process, particularly when it involves the mask mixing operation.
>
>
> We thank the reviewer for raising this point. We agree that the latent optimization in Algorithm 3 is non-convex, and that its convergence behavior should be characterized more explicitly. Importantly, this optimization is only used for downstream inverse / reconstruction tasks; the unconditional generation results remain one-step at inference time.
>
> To address the reviewer’s concern, we here add (i) convergence table of the optimization objective over the 80 iterations for Darcy Forward Problem, and (ii) Error variance across multiple random initializations for representative Darcy Forward/Inverse/Reconstruction settings. On the PDE benchmarks, we find that the optimization is empirically stable and the variance across random starts is small, and we will report these statistics in the revision.
>
> |Iter|10|20|40|60|80|
> |-|-|-|-|-|-|
> |PDE|0.727|0.173|0.152|0.149|0.145|
> |MSE|0.119|0.012|0.003|0.002|0.002|
>
> |MSE 1e-5|Fwd|Inv|Rec|
> |-|-|-|-|
> |PIDDM-ref|8.57|9.49|8.14|
> |DiffusionPDE|46.39|54.61|42.72|
>
> Q: Can you articulate the fundamental algorithmic novelty of this work beyond augmenting a standard one-step diffusion distillation objective with a physics residual penalty?
>
> Thank you for this question. We agree that simply adding an extra penalty term is not, by itself, a new primitive. **The algorithmic novelty of PIDDM lies in where and when physics is enforced in the overall diffusion pipeline**: prior methods impose PDE constraints during iterative denoising on posterior-mean surrogates, whereas PIDDM shifts physics enforcement to a post-hoc distillation stage and applies it directly to the final generated sample. This is an algorithmic change, not just a cosmetic change of loss terms. Standard one-step distillation only compresses the teacher for speed; PIDDM uses distillation as the mechanism for converting denoising-time guidance into final-sample physics supervision with one-step inference. The distillation also enables effective and efficient PDE forward/inverse/reconstruction solving. Although the loss function is a simple combination, the empirical results show that overall diffusion algorithm reformulation is effective across multiple PDE benchmarks and downstream tasks, suggesting that diffusion distillation is a promising paradigm for future PDE-constrained generative modeling.

---

> > ### Author Rebuttal · Reviewer_ze8L · 2026-04-03
> >
> > Thank you to the authors for the detailed rebuttal and the additional experiments.
> >
> > The proposed dynamic-weighting strategy is a practical solution that effectively addresses my concerns regarding the sensitivity of the distillation loss weight. Furthermore, the new convergence and variance tables adequately resolve my questions about the stability of the latent optimization process in downstream tasks. I recommend including these new results and discussions in the final version.
> >
> > Regarding the novelty, while I understand your argument about shifting the "where and when" of the physics constraints, I still view the core contribution as a clever engineering combination rather than a fundamental theoretical breakthrough. Nevertheless, I fully acknowledge its practical effectiveness in bypassing the Jensen's Gap and significantly improving inference efficiency.
> >
> > Overall, I appreciate the clarifications and will maintain my positive score . Best of luck with your work.

---

> > > ### Author Response · Authors · 2026-04-06
> > >
> > > We thank the reviewer for recognizing the strong empirical performance of PIDDM. We also appreciate the opportunity to clarify the theoretical positioning, and agree that this aspect can be presented more clearly in the revision. Our intended focus is **PDE error on generated samples**: the central object in PIDDM is the PDE satisfaction of the **final sample**. The discussion around posterior-mean guidance is included only to explain why supervising an intermediate surrogate need not align with that endpoint quantity.
> > >
> > >
> > > Appendix B.3 is designed to explain this point in the cleanest Gaussian setting (Gaussian and MoG settings are standard analytically tractable models in diffusion literature [1,2,3]). There is a distinguished constrained direction associated with a low-rank subspace, and along that direction the score becomes extremely sharp as the diffusion noise vanishes. As a result, teacher learning or guidance is trying to optimize a difficult object in **score space**, whereas post-hoc distillation imposes the PDE penalty directly in **data space** on the endpoint sample, which directly contracts the residual of the final sample. The same endpoint mechanism extends naturally to a MoG model. We will clarify this main-text positioning and further strengthen the appendix discussion so that the scope of the theory are more explicit. We thank the reviewer again for prompting us to sharpen this aspect of the paper.
> > >
> > >
> > > [1] Learning Mixtures of Gaussians Using the DDPM Objective
> > >
> > > [2] Learning General Gaussian Mixtures with Efficient Score Matching
> > >
> > > [3] What Does Guidance Do? A Fine-Grained Analysis in a Simple Setting

---

### Official Review · Reviewer_ABVR · 2026-03-13

**Soundness:** 2
**Presentation:** 1
**Significance:** 2
**Originality:** 2
**Overall Recommendation:** 3
**Confidence:** 4

**Summary:**

This paper studies physics-informed diffusion models for PDE reconstruction. Jensen's gap is pointed out, as a mechanism explanation for the novelty of this paper. PIDDM, the post-hoc distillation framework in which a standard teacher diffusion model is trained first, then distilled into a one-step student.

**Compliance With Llm Reviewing Policy:**

Affirmed.

**Final Justification:**

The authors engaged thoroughly across two rounds with helpful empirical additions (compute-matched comparison, multi-architecture results, teacher ablation). In the second response, they clarified the intended theoretical claim more precisely: the core argument is that score-space PDE enforcement is ill-conditioned, whereas distillation places the penalty directly in data space on the endpoint sample, which is better conditioned for optimization. The extension to the mixture-of-Gaussians setting strengthens this intuition. I appreciate this clarification — it positions the contribution more precisely than the original framing. My concern about the gap between the paper's theoretical positioning and what is formally established is partially alleviated, though the main text would benefit from foregrounding this data-space-vs-score-space argument rather than the Jensen's gap framing. I maintain my score of 3 (Weak Reject), but acknowledge the paper makes a solid empirical contribution and the theoretical intuition, when stated precisely, is reasonable.

**Key Questions For Authors:**

1. Would the training process of student model trade off with a more costly guidance method on teacher model alone?

2. How sensitive are the results to teacher quality and teacher NFE at distillation data generation time?

3. What can be the application scenario that requires extreme efficiency?

**Limitations:**

There are more limitations.
1. Jensen gap only serves as motivation but is not enclosed by testing the student model.

2. Limited architecture experiment (other than FNO).

**Strengths And Weaknesses:**

Strengths:
1. Strong efficiency gains (after distillation).
2. Experiment across several PDE benchmarks and downstream tasks consistently show improved physical consistency and competitive statistical fidelity.

Weaknesses:
1. Limited theoretical justification: avoiding posterior mean mismatch does not guarantee recovery of the correct distribution after guidance.
2. The manuscript has some over claims and grammar issues.

---

> ### Author Rebuttal · Authors · 2026-03-31
>
> Q1. Limited theoretical justification: avoiding posterior mean mismatch does not guarantee recovery of the correct distribution after guidance. The manuscript has some over claims and grammar issues.
>
> We thank the reviewer for the insightful comment. We agree that avoiding the posterior-mean mismatch alone does not guarantee distribution fidelity in general. This is precisely one key motivation of PIDDM: **rather than further refining PDE guidance within the iterative denoising pipeline, PIDDM takes a different route by moving physics enforcement to the final generated samples during distillation.** Our distributional-fidelity claim is therefore empirical, supported by Tables 1–3. On the theory side, Theorem B.2 shows that under stated assuptions, when the teacher/guidance has non-zero constraint violation, the post-hoc distillation objective can systematically reduce this violation via the final-sample penalty. We will revise the paper to make this scope explicit and avoid overstating a theoretical distributional guarantee.
>
> Q2. Would the training of student trade off with more costly teacher-only guidance?
>
> We thank the reviewer for this insightful question. We agree that a more accurate guidance estimate can in principle be obtained with substantially higher computation. To clarify this trade-off, we compared PIDDM-ref against two more costly teacher-only alternatives on Darcy Forward Task: (i) D-Flow, which backpropagates the PDE loss through the full 100-step sampling trajectory, and (ii) a Monte Carlo teacher-guidance variant (MC) that uses 30 teacher samples from a 100-step DDPM sampler to better approximate the guidance expectation. Both alternatives require **backpropagation through full diffusion trajectories, making them considerably more expensive online and harder to optimize**, yet they still underperform PIDDM-ref under comparable compute budgets.
>
> |Method|Wall-clock/sample (s)|Extra train(s)|PDE 1e-4|MSE 1e-1|
> |-|-|-|-|-|
> |PIDDM-ref|1.17|~3600|0.145|0.316|
> |D-Flow|~1900|0|0.459|0.477|
> |MC| ~3900|0|0.509|0.536|
>
> Q3. Sensitivity to teacher quality and NFE
>
> We thank the reviewers for the valuable comment. Figure 3(a) already studies **teacher NFE** for distillation data generation and shows diminishing returns beyond 500 denoising steps. To further assess **teacher quality**, we add a teacher-checkpoint ablation. We find that student performance improves with teacher quality early on, but the gains become marginal once the teacher is trained beyond 6k iterations.
> ||2k|4k|6k|8k|10k|
> |-|-|-|-|-|-|
> |MMSE e-2|.174|.136|.121|.119|.112|
> |SMSE e-2|.124|.098|.085|.085|.082|
> |PDE e-4|.336|.247|.231|.229|.226|
>
> Q4. What can be the application scenario that requires extreme efficiency?
> We thank the reviewer for the insightful question. PIDDM targets repeated-query, latency-sensitive scientific workflows, such as uncertainty quantification [1], online data assimilation / reconstruction from sparse observations [2], digital twins / virtual sensing [3], and design optimization with repeated PDE evaluations [4]. In these regimes, as Reviewer 26Pc noted, one-step distillation is practically attractive because iterative sampling is expensive.
>
> Q5. Jensen gap only serves as motivation but is not enclosed by testing the student model.
>
> We thank the reviewer for raising this point. We agree that Jensen-gap mainly serves as motivation. Regarding “student validation,” we clarify that **Jensen gap is not the operative quantity for the student objective**. Jensen gap arises when PDE loss is imposed on $\mathbb{E}[x_0|x_t]$ during diffusion training or sampling, which is often formulated as $R(\mathbb{E}[x_0|x_t]) \neq \mathbb{E}[R(x_0)|x_t]$ . PIDDM does not operate in that regime: it distills a one-step generator and applies the PDE loss directly to final outputs. In particular, for the distilled student $d(\epsilon)$, the output is deterministic given $\epsilon$, and therefore
> $R(\mathbb{E}[d(\epsilon)|\epsilon])=\mathbb{E}[R(d(\epsilon))|\epsilon]=R(d(\epsilon)).$ We will clarify this in our revision.
>
> Q6. Limited architecture experiment (other than FNO).
>
> We chose FNO because it is a strong and standard backbone in PDE operator learning. We here add results on a **U-Net** and a **CNN-based auto-encoder (AE)** on the Darcy benchmark in Table.1, and show the quantitative improvement.
>
> ||PIDDM(UNet)|DiffusionPDE(UNet)|PIDDM(AE)|DiffusionPDE(AE)|
> |-|-|-|-|-|
> |MMSE e-2|0.243|0.753|0.310|0.975|
> |SMSE e-2|0.211|0.431|0.257|0.692|
> |PDE e-4|0.743|1.566|0.902|1.848|
> |FPD|0.948|1.651|1.182|1.906|
>
>
>
> [1] Derivative-Informed Neural Operator Acceleration of Geometric MCMC for Infinite-Dimensional Bayesian Inverse Problems
>
> [2] Efficient deep data assimilation with sparse observations and time-varying sensors
>
> [3] Deep neural operator-driven real-time inference to enable digital twin solutions for nuclear energy systems
>
> [4] Fourier Neural Operator with Learned Deformations for PDEs on General Geometries

---

> > ### Author Rebuttal · Reviewer_ABVR · 2026-03-31
> >
> > I thank the authors for the thorough empirical additions, which are genuinely helpful. The compute-matched comparison (Q1), teacher quality ablation (Q2), and multi-architecture results (L2, U-Net and CNN-AE) are all welcome and address the empirical breadth of the evaluation. The application scenarios (Q3) are reasonable.
> >
> > My core concern, however, remains with the theoretical positioning of the paper (W1, L1). The authors agree to soften the "theoretically sound" claim, which is appreciated, but the underlying issue persists: the paper's central motivation is bypassing the Jensen's gap, yet the theoretical justification is limited to Theorem B.2, which has narrow scope and assumptions. The authors' own argument that the Jensen gap is "not operative" for the deterministic student (since $R(E[d(\epsilon)|\epsilon]) = R(d(\epsilon))$) is technically correct but creates a tension — if the gap vanishes trivially by construction, it is unclear why it serves as the paper's key framing. The gap is presented as the core insight motivating distillation, but the resolution is architectural rather than theoretical.
> >
> > This concerns a core tenet of the work: the paper positions itself as providing a principled theoretical alternative to prior PDE-constrained diffusion methods, but the contribution is ultimately empirical (effective post-hoc distillation with PDE loss on final samples). The empirical results are solid, but the theoretical framing overstates what is formally established. Addressing this would require restructuring the narrative, which is beyond what a short rebuttal can accomplish.
> >
> > I maintain my current score.

---

> > > ### Author Response · Authors · 2026-04-06
> > >
> > > Thank you for the follow-up. We appreciate the opportunity to clarify the intended claim. Our intended focus is **PDE error on generated samples**: the central object in PIDDM is the PDE satisfaction of the final sample. The discussion around posterior-mean guidance is included only to explain why supervising an intermediate surrogate need not align with that endpoint quantity. This is also what our rebuttal additions were designed to probe: even with more costly guidance, the guided teacher still underperforms the distilled student in the compute-matched teacher-student comparison. **At a high level, teacher learning or guidance is trying to optimize a difficult object in score space, whereas distillation imposes the constraint directly in data space on the final sample.**
> > >
> > > Appendix B.3 is designed to explain exactly this point in the cleanest Gaussian setting. There is a distinguished constraint direction $e_d$ associated with a low-rank subspace. **Along that direction, the ground-truth score becomes extremely sharp as the diffusion noise vanishes,** leading to ill-conditioned learning targets for diffusion training. Thus, direct optimization and guidance on that score component can be hard to capture along this sharp direction. In contrast, we show that **post-hoc distillation places the penalty directly on the endpoint coordinate** along $e_d$, i.e. in data space, which is more benign for optimization because we mainly need to push the error along one direction to zero rather than learn an ill-conditioned target. In this way, the PDE error can be controlled more effectively.
> > >
> > > We would also want to clarify that the same mechanism and intuition can be extended to the more general mixture-of-Gaussian (MoE) setting, the one that has received extensive study in theoretical research [1,2,3]. Mathematically, we consider the following data model:
> > > $$
> > > x_0\sim \sum_{k=1}^K \pi_k \mathcal N(\mu_k,\Sigma_k),
> > > $$
> > > where $\pi_k$ is the mixture weight and $\mathcal N(\mu_k,\Sigma_k)$ is the $k$-th mode. Similar to the theory in our paper, we consider the low-rank model by setting, for all $k$,
> > > $$
> > > \mu_k=(\mu_{k,1},\dots,\mu_{k,d})^\top,\qquad
> > > \Sigma_k=\mathrm{diag}(\lambda_{k,1},\dots,\lambda_{k,d-1},0).
> > > $$
> > > Thus, for all $k$, the $k$-th component is singular in the same last coordinate, while the location along that coordinate is allowed to vary across modes. Equivalently, each component satisfies the affine hard constraint $F_k(x)=(x_d-\mu_{k,d})^2=0$. This is the direct MoG analogue of the low-rank Gaussian setting in Appendix B.3.
> > >
> > > We can then calculate the score of the diffusion marginal quantitatively and show that it is ill-conditioned for teacher learning and guidance.
> > >
> > > Following the variance-preserving setting in Appendix B.3, we have the following diffusion marginal distribution:
> > > $$
> > > p_t(x)=\sum_{k=1}^K \pi_k\phi(x;\alpha_t\mu_k,\Sigma_{k,t}),
> > > $$
> > > where $\phi(x;m,S)$ is the Gaussian density with mean $m$ and covariance $S$, and, for all $k$,
> > > $
> > > \Sigma_{k,t}:=\alpha_t^2\Sigma_k+\sigma_t^2 I.
> > > $
> > >
> > > We now calculate the score of this diffusion marginal:
> > > $$\nabla\log p_t(x)=\sum_{k=1}^K\frac{\pi_k\phi(x;\alpha_t\mu_k,\Sigma_{k,t})}{p_t(x)}\Bigl(-\Sigma_{k,t}^{-1}(x-\alpha_t\mu_k)\Bigr).
> > > $$
> > >
> > > Then we can show that the ground truth score and ill-conditioned along the constrained direction:
> > > $$
> > > \langle \nabla\log p_t(x),e_d\rangle=-\frac{1}{\sigma_t^2}\sum_{k=1}^K\frac{\pi_k\phi(x;\alpha_t\mu_k,\Sigma_{k,t})}{p_t(x)}\bigl(x_d-\alpha_t\mu_{k,d}\bigr).
> > > $$
> > > Here we show that, in our setting, the score on the constrained direction carries the same sharp factor $1/\sigma_t^2$ as in Appendix B.3. Hence, **as $t\to 0$, teacher learning and guidance still need to resolve an increasingly ill-conditioned score exactly on the PDE-relevant subspace** (i.e., $\sigma_t^2\rightarrow 0$). As a consequence, the hardness of learning such a challenging target score function would finally lead to large constraint error of the generated samples.
> > >
> > > By contrast, distillation acts directly on the endpoint in data space. The final-sample PDE penalty is imposed on the constrained coordinate itself, so **the student suppresses the violating component of the generated sample rather than resolving an increasingly singular score field along the full trajectory.** This is exactly the same high-level mechanism as in Appendix B.3, and it transfers naturally to the MoG setting as well.
> > >
> > > We hope this helps clarify our main motivation and better illustrate the theoretical analysis of final-sample constraint violation reduction under distillation in Appendix B.3. We will revise the main text to make this focus more explicit and include the MoG analysis as further empirical support for this claim.
> > >
> > > References
> > >
> > > [1] Learning Mixtures of Gaussians Using the DDPM Objective
> > >
> > > [2] Learning General Gaussian Mixtures with Efficient Score Matching
> > >
> > > [3] What Does Guidance Do? A Fine-Grained Analysis in a Simple Setting

---

### Official Review · Reviewer_26Pc · 2026-03-24

**Soundness:** 3
**Presentation:** 3
**Significance:** 3
**Originality:** 3
**Overall Recommendation:** 4
**Confidence:** 4

**Summary:**

The paper argues that existing physics-constrained diffusion methods enforce PDE constraints on the posterior mean rather than on the final generated sample, which creates a Jensen’s Gap and leads to a trade-off between physical consistency and generative fidelity. The proposed method, PIDDM, avoids this by first training a standard teacher diffusion model, then performing post-hoc distillation into a one-step student model while directly applying PDE residual losses on the final output. The method is evaluated on several PDE benchmarks, including Darcy, Poisson, and Burgers, and is claimed to outperform prior baselines in both accuracy and PDE satisfaction, while also reducing inference cost.

**Compliance With Llm Reviewing Policy:**

Affirmed.

**Key Questions For Authors:**

Can you clarify how much of PIDDM’s gain comes from distillation itself versus the fact that PDE constraints are enforced on the final sample rather than the posterior mean?
How well does the student preserve the diversity of the teacher distribution, beyond mean/std and FPD metrics?
Can you provide a more controlled compute-matched comparison against baselines, especially methods with optional refinement or optimization steps?

**Limitations:**

Same as the previous question

**Strengths And Weaknesses:**

Strength
Clear motivation and central insight. The Jensen’s Gap framing is intuitive and gives a strong conceptual reason why prior posterior-mean-based PDE guidance may be suboptimal.
Simple and practical method. The approach is post-hoc, modular, and does not require redesigning diffusion training itself.
Efficiency advantage. Distillation to a one-step student is practically appealing, especially for scientific generation tasks where iterative sampling is expensive.
Broad task coverage. The method is presented not only for unconditional generation, but also for forward, inverse, and reconstruction setting

Weakness
Heavy dependence on teacher quality. The method inherits performance limits from the pretrained teacher, which the paper itself acknowledges as a limitation.
Claim of “theoretically sound” feels somewhat overstated. The argument for bypassing Jensen’s Gap is convincing at a high level, but the core evidence in the main paper is still mostly empirical rather than a full end-to-end theoretical guarantee.
Comparison fairness could be discussed more carefully. PIDDM benefits from distillation and one-step inference, while some baselines operate under different computational or optimization regimes. It would help to standardize budgets and settings more explicitly

---

> ### Author Rebuttal · Authors · 2026-03-31
>
> Q1. Claim of “theoretically sound” feels somewhat overstated. The argument for bypassing Jensen’s Gap is convincing at a high level, but the core evidence in the main paper is still mostly empirical rather than a full end-to-end theoretical guarantee.
>
> We thank the reviewer for this important comment. We agree that the phrase “theoretically sound” may be read as claiming a full end-to-end theorem-level guarantee, which was not our intended meaning, and we will replace it with a more precise formulation. Our intended claim is specific: PIDDM removes the Jensen-gap-related mean-vs-sample mismatch discussed in the paper by construction. In prior physics-guided diffusion methods, PDE constraints are imposed on intermediate mean-based denoising surrogates, which need not align with the physical consistency of the realized final sample. In contrast, PIDDM applies the PDE loss directly to the final output of the distilled one-step student, so this particular mismatch does not arise in the student objective. Appendix B.3 further provides a theorem showing that, under the stated assumptions, post-hoc distillation with a final-sample penalty can reduce constraint violation even when the pretrained teacher has non-zero violation. We will revise the wording in the main text to reflect this precise scope more accurately.
>
> Q2. Comparison fairness could be discussed more carefully. Can you provide a more controlled compute-matched comparison against baselines, especially methods with optional refinement or optimization steps?
>
> We thank the reviewer for this important comment and agree. Here, we use matched online wall-clock time per sample as the primary compute budget, measured on the same hardware for all methods. Alongside wall-clock (second), we will also report NFE, the number of full-Network Backward Passes (NBP), so that hidden inference-time costs are made explicit. We will then provide a controlled compute-matched comparison on a Darcy benchmark  in Table. 1 by tuning the optional refinement/optimization budgets of each baseline to comparable online cost.
> ||PIDDM-1|PIDDM-ref|DiffusionPDE|PIDM|DFlow|ECI|
> |-|-|-|-|-|-|-|
> |time|0.00427|1.17|1.25|1.281|19.302|1.294|
> |NFE|1|80|80|300|2000|300|
> |NBP|0|80|80|0|2000|0|
> |PDE(1e-4)|0.226|0.148|1.129|1.274|0.885|1.106|
> |FPD|0.754|0.385|1.452|2.018|1.252|0.837|
> |MMSE(1e-2)|0.112|0.037|0.431|0.522|0.254|0.138|
> |SMSE(1e-2)|0.082|0.002|0.173|0.374|0.209|0.112|
>
> We observe that under a compute budget matched to PIDDM-ref, the baseline methods show only limited improvement. In particular, D-Flow becomes difficult to tune to produce meaningful PDE samples under the same online budget.
>
> Q3. Can you clarify how much of PIDDM’s gain comes from distillation itself versus the fact that PDE constraints are enforced on the final sample rather than the posterior mean?
>
> We thank the reviewer for this important question. The gain of PIDDM comes from the combination of one-step distillation and final-sample PDE supervision, and **these two components play different roles**.
> - Distillation alone mainly provides the one-step speed-up, but does not by itself improve PDE satisfaction.
> - Final-sample PDE supervision is the key ingredient for improving physical consistency.
> - PIDDM combines both, i.e., it imposes PDE on one-step final sample, enabling efficient and effective forward/inverse/reconstruction and refinement in Sec. 3.3.
>
> To isolate these effects, we added two ablations on Darcy in Table. 1:
> - Distill = pure distillation without PDE loss;
> - D-Flow = final-sample PDE loss without distillation.
>  The results are:
>
> ||PIDDM-ref|Distill|Teacher|DFlow|
> |-|-|-|-|-|
> |MMSE(1e-2)|0.037|0.121|0.108|0.129|
> |MSME(1e-2)|0.002|0.102|0.069|0.085|
> |PDE(1e-4)|0.148|1.642|1.585|0.532|
> |FPD|0.385|0.825|0.782|0.995|
> |NFE|80|1|100|5000|
>
> These results show that distillation alone provides the efficiency gain but does not improve PDE satisfaction, whereas final-sample PDE supervision alone reduces the PDE residual at much higher cost. PIDDM combines both, achieving a better quality&compute.
>
> Q4: How well does the student preserve the diversity of the teacher distribution, beyond mean/std and FPD metrics?
>
> We thank the reviewer for the insightful question. We here provide two additional metrics using sliced wessertain distance (SWD) and average pairwise distance difference (APDD), calculated as:
>
> Given $N$ samples $x^{(1)}, \ldots, x^{(N)}$, we define $\mathrm{APD}=\frac{2}{N(N-1)}\sum_{i<j}\|x_i - x_j\|_2.$
>
> We compute this quantity separately for teacher and student samples:
> $
> \mathrm{APD}_T, \qquad \mathrm{APD}_S.
> $
>
> To quantify diversity preservation, we report  the relative error
> $
> \frac{|\mathrm{APD}_S - \mathrm{APD}_T|}{\mathrm{APD}_T}.
> $
> | |PIDDM-1|PIDDM-ref|Distill.|Teacher|
> |-|-|-|-|-|
> |SWD |0.058 |0.032 |0.103 |0.089 |
> |APDD |0.050 |0.032 |0.067 |0.034 |
>
> These results show that PIDDM preserves the teacher distribution’s diversity better than vanilla distillation, with PIDDM-ref performing best.

---

### Decision · Program_Chairs · 2026-04-30

**Decision:**

Accept (regular)

**Comment:**

The paper proposes a simple and effective reformulation of physics-informed diffusion modeling: instead of enforcing PDE constraints during iterative denoising, it applies them at the final output of a post-hoc distilled one-step student. Reviewers generally agree that this is a practically meaningful idea with strong efficiency benefits, and that the empirical results are solid across multiple PDE benchmarks and downstream tasks. The rebuttal was strong and addressed most technical concerns through additional compute-matched comparisons, ablations clarifying the role of final-sample PDE supervision, teacher-quality and architecture analyses, and discussion of weighting and optimization stability. One initially negative reviewer raised their score from 3 to 4 after discussion, leaving the overall reviewer balance positive though still somewhat borderline.

The main remaining concern is about framing rather than empirical validity. In particular, some reviewers felt that the theoretical positioning around Jensen’s gap and theoretical soundness overstated what is formally established. I agree that the contribution is best understood as an empirical and methodological advance rather than a theoretical one, and the final version should revise the presentation accordingly. That said, I do not view this issue as sufficient to outweigh the practical value of the method and the strength of the empirical evidence.